# Ionic Liquid Catalyzed Hydrolysis of Sugarcane Cellulose to Produce Reducing Sugar

Ruihuan Liu [1], Jiying Li [2], Enming Liu [1], Ahmad Ali [1], Zicheng Li [1,*] and Shun Yao [1,*]

1   School of Chemical Engineering, Sichuan University, Chengdu 610065, China; 2023223075212@stu.scu.edu.cn (R.L.); liuenming_scu@126.com (E.L.); ahmadali9849587@gmail.com (A.A.)
2   School of Medicine and Nursing, Changjiang Polytechnic, Wuhan 430074, China; lijiying686868@126.com
*   Correspondence: sculzc@scu.edu.cn (Z.L.); cusack@scu.edu.cn (S.Y.); Tel./Fax: +86-028-85405221 (S.Y.)

**Abstract:** As the most abundant bioenergy raw material in nature, cellulose can be converted into sugar by hydrolysis, which can be further degraded to produce downstream chemicals, such as polyols. Hydrolysis technology is one of the key steps in the development and utilization of cellulosic biomass resources. In this study, the ionic liquid (IL)-catalyzed hydrolysis of sugarcane cellulose into reducing sugar was studied. Firstly, the hydrolysis of sugarcane cellulose in different ionic liquids (including benzothiazolomethane sulfonate, $[HBth][CH_3SO_3]$ and 1-methyl-3-(3-sulfopropyl)-imidazolium hydrogen sulfate, $[C_3SO_3Hmim]HSO_4$) in heterogeneous and homogeneous systems to produce reducing sugar was studied. In a homogeneous system, the catalytic effect of an ionic liquid on sugarcane cellulose was explored. The pretreatment, IL dosage (0.1~1.0 g), reaction temperature (100~180 °C), addition of water (0~500 μL), and time (1~6 h) were all discovered as key conditions for hydrolysis. The acidity of an acidic ionic liquid is a key factor affecting the hydrolysis of sugarcane cellulose; meanwhile, effective pretreatment and water are also important. As a comparison, the catalytic effect of $[C_3SO_3Hmim]HSO_4$ in heterogeneous systems (the maximum yield of 5.98% for total reducing sugars, TRS) was not as good as that of $[HBth][CH_3SO_3]$ in homogeneous systems (33.97%). A higher temperature does not necessarily lead to an increased TRS yield, but it will make the maximum TRS appear earlier. At last, 732 cationic ion exchange resin was used to investigate the separation of reducing sugar and ionic liquid, and the recovery of ionic liquid was investigated by an adsorption–desorption experiment. The ionic liquid can be well separated from TRS in the $[HBth][CH_3SO_3]$ and reused at least five times.

**Keywords:** sugarcane bagasse; cellulose; ionic liquids; hydrolysis; catalysis

## 1. Introduction

With the rapid development of modern industry, people's demand for fossil fuels is increasing day by day. However, in today's shortage of non-renewable energy, such as fossil fuels, the development of renewable biomass resources is of great significance for sustainable development [1]. However, the large number of H-bonds and the effective surface area of cellulose affect its hydrolysis results. At present, the main methods for cellulose hydrolysis to sugar include traditional acid hydrolysis [2,3], solid acid hydrolysis [4–6], enzyme hydrolysis [7,8], supercritical or subcritical hydrolysis [9,10]. They all have their own inherent advantages and disadvantages, and exploring mild, green, and efficient degradation technologies has always been a common goal in both academia and industry. In recent years, with the development of green chemistry and precise synthesis, organic molecular catalysis research has become a hot field in organic chemistry [11]. Ionic liquid has the advantages of good solubility, stability, and easy recovery, which makes it important in the field of green solvents and green catalysts [12]. Ionic liquids also have unique properties, such as low vapor pressure, tunable solvation behavior, and a wide liquid range. Through structural design, different functional groups are modulated and

introduced into the negative and cation structures of ionic liquids to prepare functional ionic liquids, which can achieve specific catalytic functions [13].

Ionic liquids (ILs) are a kind of new green solvent with good designability, recyclability, and multifunction that have the advantages of low vapor pressure, tunable solvation behaviors, a wide liquid range, and dissolution scope. Therefore, their applications in the field of biomass are becoming increasingly widespread [14–17]. As a powerful and efficient cellulose solvent, the emergence of ionic liquids undoubtedly provides a new direction for the hydrolysis of cellulose and expands the current cellulose hydrolysis technology. And there are more and more reports on this research. In detail, Zhang et al. found that solid acid (NKC-9 resin)-catalyzed hydrolysis of cellulose in ionic liquid was greatly promoted by microwave heating, which resulted in a final yield of 37% [18]. 1-Butyl-3-methylimidazolium chloride ([Bmim]Cl) has been proven to facilitate the hydrolysis of cellulose with dramatically accelerated reaction rates at 100 °C in liquid acid-catalyzed systems [19] and DuPont$^{TM}$ AmberLyst$^{TM}$ 15DRY polymeric catalysts [20]. Moreover, 1-(4-sulfonic acid) butyl-3-methylimidazolium hydrogen sulfate and a catalytic amount of $MnCl_2$ could hydrolyze the microcrystalline cellulose into 5-hydroxymethylfurfural and furfural under atmospheric pressure within 300 min at 150 °C [21]. However, most of these studies were based on the addition of extra catalysts to ionic liquid systems, such as liquid acids or solid acids, and little literature reported the influence of major factors on cellulose hydrolysis in ionic liquid bodies of pure products. In fact, hydrolysis of cellulose to sugar or further degradation products can still be achieved in pure ionic liquid systems without the addition of any other catalyst, although the yield is relatively low compared to the addition of catalysts, indicating that ionic liquids are not only effective solvents for cellulose but also catalysts. Therefore, it is believed that studying the hydrolysis of cellulose in a pure ionic liquid system is an important basic work that provides a reference for the development of more effective solvation-catalysts in the future. In addition, reducing sugar, the hydrolysis product of cellulose, is miscible with ionic liquid, and the two cannot be effectively separated, and there are few related reports [22]. Therefore, it is of great significance to explore the separation of reducing sugar and ionic liquid in the actual hydrolysis system of cellulose.

Sugarcane bagasse is the main solid by-product of the sugarcane sugar industry. According to the statistics of the Food and Agriculture Organization of the United Nations in 2019, the world sugarcane bagasse production reaches 270 million tons, accounting for 15% of the total sugarcane production, and the China sugarcane planting industry produces about 20 million tons of bagasse every year [23,24]. With bagasse as a raw material, the development and production of animal feed [25], biomass board [26] and green pulp and paper [27] have shown broad application prospects. Similar to other woody biomass, the basic components of bagasse include cellulose, hemicelluloses, and lignin, which can be hydrolyzed to produce fermentable sugars, which can be used as raw materials for fermentation to produce ethanol, lactic acid, and other bulk products [28,29]. The common methods for hydrolyzing bagasse into fermentable sugar include the chemical method and the enzymatic method. Among them, the chemical method mainly includes the acid method, the alkali method, and the ionic liquid method. Due to its low price and good hydrolysis effect, sulfuric acid is often used in acid hydrolysis, but the furfural and other substances generated during acid hydrolysis can inhibit the subsequent fermentation process [30]. Alkali hydrolysis of bagasse can cause expansion, chemical bond breakage, and pore formation within the fiber of sugarcane residue, which is conducive to subsequent hydrolysis [31]. However, there are many problems in the direct preparation of fermentable sugar by acid and alkali methods, such as the low conversion rate of raw materials and the further purification of the fermentable sugar solution.

Currently, most ILs used in biomass fields belong to the imidazolium type, so more ILs with other types need to be paid enough attention to; meanwhile, the comparison of different hydrolysis modes is meaningful. Considering benzothiazolium ILs have been applied as new effective Brønsted-acidic catalysts [32] and imidazolium IL-grafted hetero-

geneous catalysts have proved successful in cellulose hydrolysis [33], it is worth exploring if the former is capable of handling the task of hydrolyzing sugarcane cellulose and the latter can be directly employed in heterogeneous catalysis. Under the aforementioned background, the preparation of reducing sugar by hydrolysis of sugarcane cellulose in the prescreened benzothiazolium ionic liquid of [HBth][CH$_3$SO$_3$] in homogeneous systems was firstly studied, and the influence trend of the main factors on the catalytic results was investigated. The structure of cellulose products before and after hydrolysis was studied by Fourier transform infrared spectroscopy (FT-IR), scanning electron microscopy (SEM), and thermogravimetric analysis (TGA) techniques. As a comparison, the catalytic effect of [C$_3$SO$_3$Hmim]HSO$_4$ in heterogeneous systems was also used for cellulose hydrolysis, and the potential deep hydrolysis product of total reducing sugars was explored by gas chromatography. Finally, the separation between the product reducing sugar and the ionic liquids was achieved by ion exchange resin, which aimed to realize the recyclability of these green solvents. The recovery of the ionic liquid was investigated by an adsorption–desorption experiment, and then its reuse performance was focused on. It is expected to provide the foundation for the large-scale application of related processes.

## 2. Materials and Methods

### 2.1. Reagents and Instruments

The reagents used in this study were sodium hydroxide (AR), hydrogen peroxide (AR), benzothiazole (98%), methane sulfonic acid (AR), *n*-methylimidazole (98%), 1,3-propanesulfonolactone (AR), sulfuric acid (AR), hydrochloric acid (AR), 1-chlorobutane (AR), chloropropene (AR), bromo-butane (AR), bromoethane (AR), bromo-hexane (AR), toluene (AR), ethyl ether (AR), phenol acetate (AR), ethyl alcohol (AR), 732H type resin (0.4~0.6 mm, exchange capacity $\geq$ 4.50 mmol/g, wet true density: 1.25~1.29 g/mL), which were all obtained by Saibole Reagent Co., Ltd., (Chengdu, China). The sample of sugarcane bagasse was collected from the local sugarcane market. It was cleaned, dried, and ground into powders with 20 mesh for further use.

The SHZ-D (III) circulating water vacuum pump was provided by the Yingyu Yuhua instrument factory (Gong Yi, China). The 101-2AB electric blast drying oven was provided by Tianjin Test Instrument Co., Ltd. (Tianjin, China). The GC-2014 gas chromatography (GC) was provided by Shimadzu Inc. (Kyoto, Japan). The KQ2200DE ultrasonic oscillator was provided by Kunshan Ultrasonic Instrument Co., Ltd. (Kunshan, China). Fourier transform infrared (FT-IR) spectra were carried out on a Spectrum Two Li600300 spectrometer (Perkin-Elmer, Waltham, MA, USA) in the spectral range between 400 and 4000 cm$^{-1}$. The AOE380 ultraviolet spectrometer (Aoyi Instruments Co., Ltd., Shanghai, China) was used for the analysis of sample absorbance. The UPH-I-10T series ultrapure water producer was provided by ULUPURE Technology Co., Ltd. (Chengdu, China). The JSM-7500F scanning electron microscope was provided by Japan Electronics Co., Ltd. (Tokyo, Japan). The TG209-F3 Tarsus thermogravimetric analyzer was provided by NETZSCH Instrument Inc. (Selb, Germany).

### 2.2. Preparation of Ionic Liquids

After the screening by pilot experiments, the ILs of [HBth][CH$_3$SO$_3$] and [C$_3$SO$_3$Hmim] HSO$_4$ were selected as the ideal catalysts for the homogeneous and heterogeneous systems in the following experiments, respectively. The synthesis method of ionic liquid benzothiazolomethane sulfonate (abbreviated as [HBth][CH$_3$SO$_3$]) was referred to in our previous research results [34], and the synthesis route is shown in Figure 1a. The specific experimental steps are as follows: 0.1 mol benzothiazole was weighed and dissolved in 50 mL anhydrous ethanol, which was added and placed in the flask in an ice bath, 0.11 mol of methanesulfonic acid was weighed and dissolved in 20 mL of water. It was slowly added to the flask containing benzothiazole under full agitation, and the mixture reacted at room temperature for more than 4 h after a drip. After the reaction, the water and ethanol were removed by vacuum distillation to obtain the crude product. The crude product was

washed with 10 mL ethyl acetate for 3 to 5 times, then recrystallized in anhydrous ethanol for 2 to 3 times, and finally colorless flake crystals were obtained, which were dried under vacuum for further use.

**Figure 1.** The synthesis routes of (**a**) [HBth][CH$_3$SO$_3$] and (**b**) [C$_3$SO$_3$Hmim]HSO$_4$.

As for 1-methyl-3-(3-sulfopropyl)-imidazolium hydrogen sulfate ([C$_3$SO$_3$Hmim]HSO$_4$) in Figure 1b, 1 mol of 1,3-propanesulfonic acid lactone was added to the flask containing toluene, and under ice bath conditions, an equal amount of N-methylimidazole was slowly added dropwise. Then the entire system was stirred and reacted at room temperature for 2 h. After filtration, the resulting white solid was washed with ether, dried, and ready for use. A certain amount of this intermediate product was dissolved in water, and an equal molar amount of diluted sulfuric acid was mixed with it slowly at room temperature. After stirring under reflux at 90 °C for 2 h, the final product of [C$_3$SO$_3$Hmim]HSO$_4$ could be obtained, and then further vacuum distillation and drying were performed. The commonly used ILs of [Bmim]Cl, [Amim]Cl, [Bmim]Br, [Emim]Br, and [Prmim]Br were all directly provided by TCI Company (Shanghai, China).

*2.3. Hydrolysis of Sugarcane Cellulose by Ionic Liquids*

The raw material of sugarcane bagasse (20 mesh) was mixed with distilled water in a solid–liquid ratio of 1:20 (*w/v*); after stirring for 3 h at 60 °C and filtering, the raw material was dried at 80 °C and ground to 40~100 mesh. Then the powders were mixed with a 7% NaOH aqueous solution in a solid–liquid ratio of 1:40 (*w/v*). After stirring at 70 °C for 8 h and filtering, the residue was collected and washed with distilled water to neutralize it to obtain sugarcane cellulose, which was dried for further hydrolysis. Here Figure 2 summarizes the whole contents of the pretreatment and hydrolysis study with ILs in various modes, and the experimental details are introduced in the following sections.

2.3.1. Homogeneous System Hydrolysis

In a homogeneous reaction system, sugarcane cellulose was pretreated to break the β-1, 4-glucoside bond. Sugarcane cellulose existed mostly in molecular form in the system, partially or completely dissolved in an ionic liquid. In this study, the imidazolium IL of [Bmim]Cl was first used to pretreat sugarcane cellulose because it is the most frequently used IL for such a task, and the differences between other ionic liquid pretreatments were compared. The specific experimental methods were as follows: 50 mg of sugarcane cellulose was added into a flask containing 2 g [Bmim]Cl, and after pretreatment at 100 °C by stirring for 30 min, the mixture system composed of sugarcane cellulose and ionic liquid formed a relatively uniform and transparent liquid. A certain amount of [C$_3$SO$_3$Hmim]HSO$_4$ ionic liquid (0.1~1.0 g) and a small amount of water (50~500 μL) were added to the pretreated sugarcane cellulose ionic liquid solution, and then the reaction was started at a certain temperature (100~180 °C). Every once in a while, a certain quality of sample was taken out and placed in a centrifugal tube, and a small amount of water was mixed with thorough shaking. After centrifugation, the supernatant was collected for analysis. The whole process lasted 1~3 h.

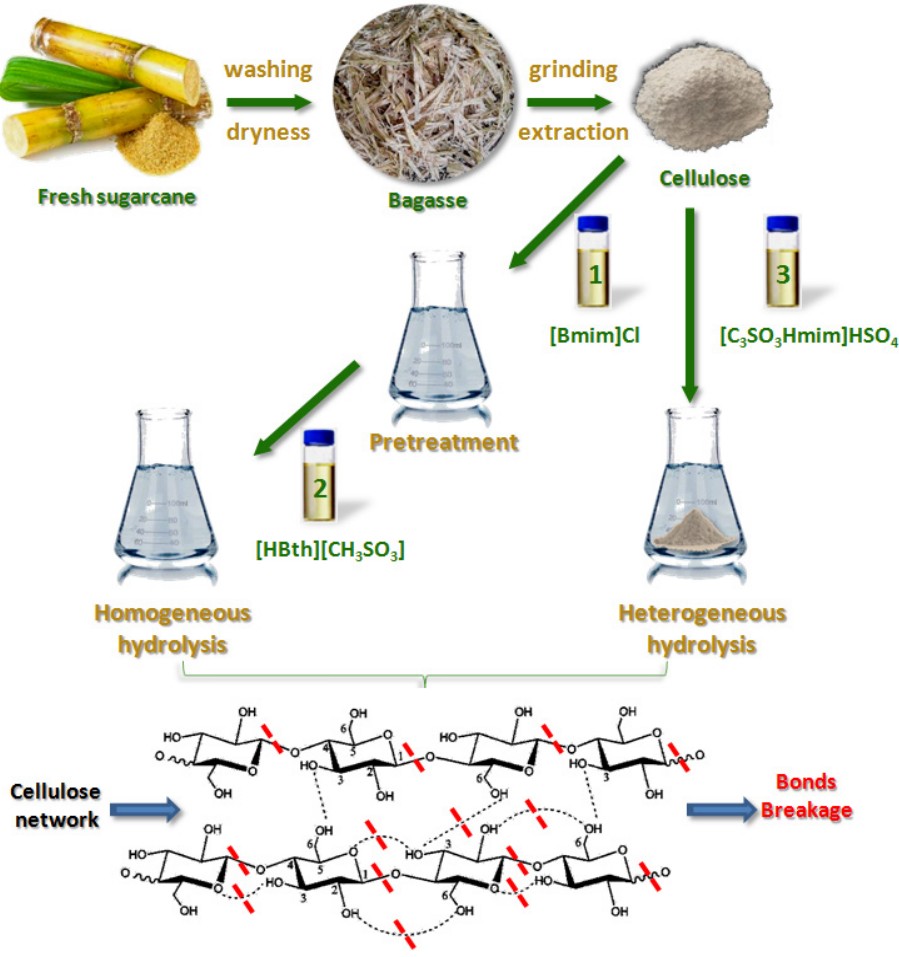

**Figure 2.** The scheme of the whole IL-catalyzed cellulose hydrolysis of sugarcane bagasse (green arrow: diagram of the process of extracting cellulose from sugarcane and hydrolyzing it; blue arrow: the process of breaking the chemical bonds of cellulose network; yellow font: experimental operations; green font: raw materials and ILs).

2.3.2. Heterogeneous System Hydrolysis

In the heterogeneous system, sugarcane cellulose was not pretreated by the above IL, which existed in solid form in the system and was not dissolved in the system. The specific experimental process was described as follows: Sugarcane cellulose was added to an aqueous solution containing 2 mL of ionic liquid with a certain concentration at a set temperature and stirred for a period of time. After the reaction, the hydrolyzed sample was transferred to a centrifuge tube and immediately quenched with ice water. After centrifugation, the supernatant was taken for analysis.

*2.4. Determination of Reducing Sugar in Hydrolysis Product*

The content of TRS was determined by the phenol-concentrated sulfuric acid method [35]. A total of 0.1 mL of the supernatant was taken to be measured and diluted to a certain multiple, then a pipet was used to measure a certain volume of the diluted liquid to be measured in a 25 mL colorimetric tube and mixed with water to 2 mL. Only 2 mL of water was added into one of the colorimetric tubes as the reference solution. A total of 1 mL of freshly prepared 5% phenol solution was added and shaken well, then 5 mL of concentrated $H_2SO_4$ was added and shaken fully. After cooling at room temperature for 20 min, the color of the system turned orange, and the absorbance at 490 nm was determined. The absorbance value remained between 0.1 and 0.3. The concentration and yield of TRS were calculated according to the standard curve of glucose.

### 2.5. Development of Standard Curves for Related Objectives

For the quantitation required in the following experiments, the standard curves of glucose, [Bmim]Cl, [$C_3SO_3$Hmim]$HSO_4$, and [HBth][$CH_3SO_3$] were developed according to the developed methods [36–39], which were summarized in Figures S1–S4 as well as experimental details in Supplementary Information (SI).

### 2.6. Characterization of Sugarcane Cellulose before and after Hydrolysis

#### 2.6.1. Infrared (IR) Spectroscopic Analysis

The sugarcane cellulose samples before and after hydrolysis were compressed by KBr. The structure of the samples was analyzed by a Fourier transform infrared spectrometer using the transmittance analysis mode. The spectral scanning range was 4000~400 cm$^{-1}$.

#### 2.6.2. SEM Analysis

The sugarcane cellulose samples before and after hydrolysis were frozen in liquid nitrogen and sprayed with gold on the surface. The surface morphology was observed by a JSM-7500F scanning electron microscope at an accelerated voltage of 5.0~10.0 kV.

#### 2.6.3. Thermogravimetric (TGA) Analysis

The thermal properties of sugarcane cellulose samples before and after hydrolysis were tested by a thermogravimetric analyzer using static TGA mode from 30 °C to 800 °C in an $N_2$ atmosphere with a heating rate of 10 °C/min and a nitrogen flow rate of 30 mL/min.

### 2.7. Separation of Ionic Liquids from the Hydrolysis Product in Post-Treatment

After hydrolysis, 25 mL of diluted supernatant was mixed with 1 g (dry weight) of 732H type resin and shaken at 30 °C until adsorption equilibrium. When the adsorption was completed, the absorbance of the ionic liquid in the filtrate was determined by UV, and the absorbance of the ionic liquid and TRS was determined by UV after the color was developed by the phenol-sulfuric acid method. The concentration of the adsorbed ionic liquid and TRS in the solution was calculated by the standard curve, and the adsorption rate of $E(\%)$ was calculated according to Formula (1) to investigate the separation of the ionic liquid and reducing sugar as follows:

$$E\,(\%) = \frac{(C_0 - C_1) \times V}{C_0 \times V} \times 100\% = \frac{(C_0 - C_1)}{C_0} \times 100\% \tag{1}$$

where $C_0$ is the concentration of ionic liquid, or TRS, in the solution before adsorption (μg/mL). $C_1$ is the concentration of ionic liquid, or TRS, in the adsorbed solution (μg/mL). $V$ is the volume of the sample solution (mL).

The static adsorption saturated 732H resin was placed in a 100 mL conical bottle, 25 mL of HCl aqueous solution with a certain concentration was added, and the resin was shaken on a shaking table until desorption was complete. Then the concentration of ionic liquid in the filtrate was determined, and the desorption rate ($D$, %) was calculated according to Formula (2) as follows:

$$D\,(\%) = \frac{C_2 \times V}{(C_0 - C_1) \times V} \times 100\% = \frac{C_2}{(C_0 - C_1)} \times 100\% \tag{2}$$

where $C_2$ is the concentration of ionic liquid in the solution after analysis; $C_1$ and $C_0$ are the same; $V$ is the initial volume of the sample solution and the volume of the analytical solution, respectively; and because the volume of the two is the same, it is uniformly represented by $V$.

## 3. Results and Discussion

### 3.1. Homogeneous Catalytic Hydrolysis with [HBth][CH₃SO₃]

The hydrolysis of cellulose is actually the process of breaking the beta-1, 4-glucoside bond. Only by breaking the β-1, 4-glucoside bond can the subsequent catalytic conversion reaction be carried out, which is a crucial step. By using some pretreatment methods, the degree of polymerization can be reduced and the effective surface area can be increased, making the hydrolysis reaction of cellulose easier to carry out. In this section, ionic liquid [Bmim]Cl was first used for the pretreatment of sugarcane cellulose. The free $Cl^-$ and $[Bmim]^+$ ions in the ionic liquid interact with H and O atoms in the cellulose hydroxyl groups, respectively. Due to the strong electronegativity of $Cl^-$, its strong traction effect towards H atoms greatly weakens both the intermolecular and intramolecular hydrogen bonds within cellulose structures. When the charge of hydroxyl groups is dispersed to a high degree, the aggregated networks of cellulose will be disrupted, and then the molecular chains will be broken [40]. Most sugarcane cellulose exists in a molecular form in pretreatment systems and completely dissolves in the ionic liquid. In order to understand the degradation of sugarcane cellulose in pretreatment, reducing sugar was detected in the sugarcane cellulose solution that had only been pretreated, and it was found that the yield of hydrolyzed reducing sugar was only 0.75% because of the mild conditions. It can be seen that the hydrolysis reaction hardly occurs during the pretreatment of sugarcane cellulose, so the contribution to the subsequent reduction in sugar yield can be ignored. After pretreatment, sugarcane cellulose is completely or partially dissolved in the ionic liquid. When acidic ionic liquid catalysts are added to the system, sugarcane cellulose and acidic ionic liquid will form a nearly homogeneous environment.

### 3.1.1. Effect of Catalyst Dosage on Yield of Reducing Sugar

The influence of catalyst amount on TRS yield was explored by changing the dosage of catalyst [HBth][CH₃SO₃]. The specific experimental results are shown in Figure 3. It can be found that when the dosage of [HBth][CH₃SO₃] was 0.1 g, the yield of TRS produced by sugarcane cellulose hydrolysis for 1 h was only 1.51%. If the dosage of [HBth][CH₃SO₃] increased to 0.2 g, when the dosage of [HBth][CH₃SO₃] continued to increase, the yield of TRS showed a gradual downward trend. It can also be deduced from the figure that when the dosage of catalyst was small, the yield of TRS basically showed an upward trend with the extension of time. With the increase in the amount of catalyst, the TRS yield showed a decreasing trend with the extension of time. When the concentration of ionic liquids increases, the viscosity of the entire reaction system rises, affecting its permeation in cellulose as well as heat and mass transfer. Usually, it takes 3~5 h to hydrolyze cellulose using 72% sulfuric acid or 42% hydrochloric acid. Compared to the process explored in this study, strong inorganic acids that are not environmentally and personnel friendly are used, and this situation needs to be improved.

### 3.1.2. Effect of Reaction Temperature on Yield of Reducing Sugar

For a chemical reaction process, temperature is a very important factor; similarly, temperature is also crucial for the hydrolysis of sugar cane cellulose to produce TRS and further hydrolysis of sugar. The hydrolysis temperatures were investigated at 100 °C, 120 °C, 140 °C, 150 °C, 160 °C, and 180 °C. Figure 4 shows the variation in TRS yield from sugarcane cellulose hydrolysis with time when the amount of catalyst [HBth][CH₃SO₃] is 0.2 g. It can be seen from the figure that when the temperature rises from 100 to 120 °C, the yield of TRS increases gradually from 10.58% to 28.35% in 1 h. This is because with the increase in temperature, the molecular movement intensifies and the viscosity of the system decreases, thus accelerating the mass transfer movement and making it easier for $H^+$ to approach the sugarcane cellulose substrate to participate in the hydrolysis reaction. The increase in temperature enhanced the ionization of $H_2O$, which increased the concentration of $H^+$ in the solution to a certain extent, thus accelerating the hydrolysis of sugarcane cellulose to TRS. However, TRS showed a decreasing trend as the temperature

continued to rise to 180 °C. This is also consistent with the previous hydrolysis behaviors of sugarcane cellulose in $[C_3SO_3Hmim]HSO_4$, because the hydrolysis of sugarcane cellulose is a continuous process, and too high temperature is not conducive to the existence of TRS, and more degradation occurs, so the yield decreases. In addition, in the short studied time range, the content of TRS produced by sugarcane cellulose hydrolysis increased with the extension of time at low temperatures. However, with the increase in temperature and the extension of time, the TRS content began to decrease again, which was consistent with the hydrolysis law of sugarcane cellulose in $[C_3SO_3Hmim]HSO_4$.

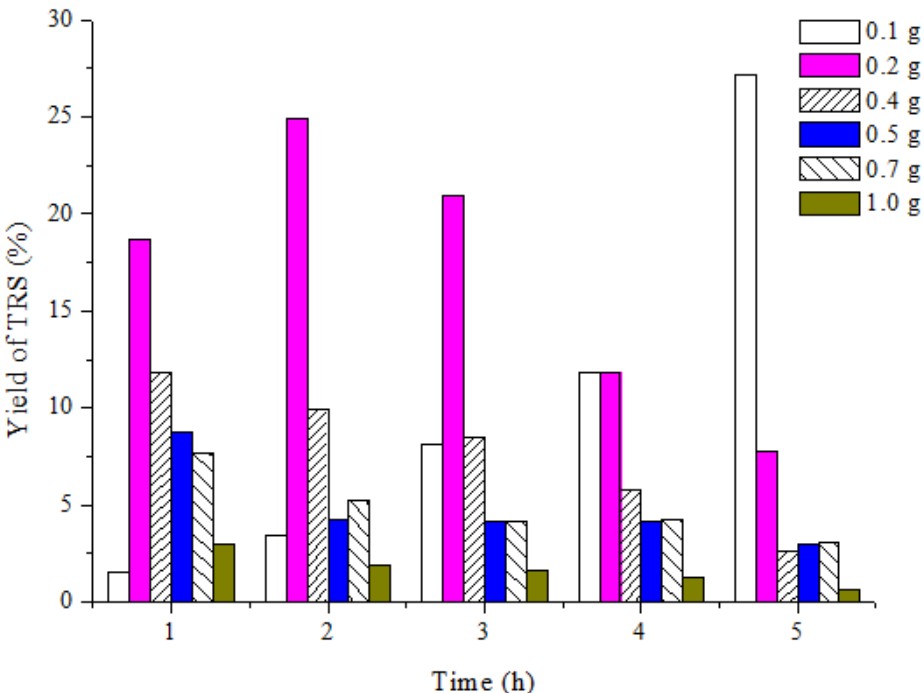

**Figure 3.** Effect of $[HBth][CH_3SO_3]$ dosage on the bagasse cellulose conversion into TRS.

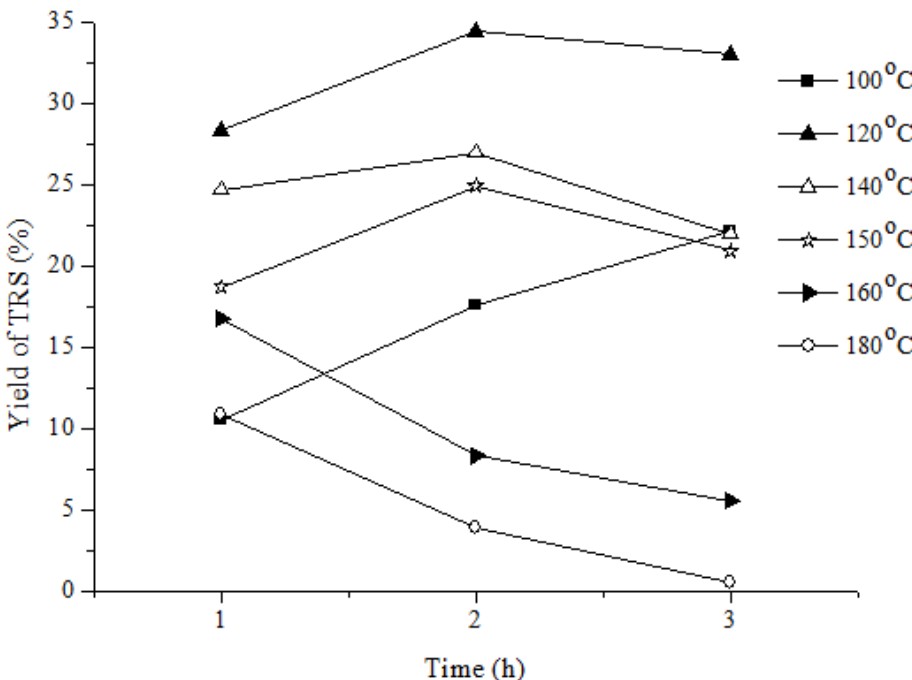

**Figure 4.** Effect of reaction temperature on the bagasse cellulose conversion into TRS.

### 3.1.3. Comparison on [HBth][CH$_3$SO$_3$] and Water

In order to discover the contribution of ionic liquid on the hydrolysis of sugarcane cellulose to prepare TRS, pure water was used as a reference to investigate the hydrolysis of sugarcane cellulose in the system with only catalyst or only water. The results are shown in Table 1. When only acidic ionic liquid catalysts of [HBth][CH$_3$SO$_3$] are added to the pretreated sugarcane cellulose ionic liquid system without adding additional water, i.e., only a small amount of strongly bound water is carried by the ionic liquid itself in the system, the yield of TRS is only 9.50%. Correspondingly, adding additional water to the pretreated sugarcane cellulose ionic liquid system without adding acidic catalysts resulted in a TRS yield of only 6.16%. If no acidic ionic liquid catalyst or water is added to the pretreated sugarcane cellulose ionic liquid system, and the pretreatment system is directly reacted under the same conditions, the analysis shows that the yield of TRS is only 1.78%. The results indicate that both H$^+$ and H$_2$O, as reactants in the initial stage of the reaction, are important for the hydrolysis reaction of sugarcane cellulose. If lacking, it will seriously limit the hydrolysis reaction of sugarcane cellulose.

**Table 1.** The comparison of the effects of catalyst and water on hydrolysis (IL: [HBth][CH$_3$SO$_3$], TRS: total reducing sugars).

| No. | H$_2$O Volume (μL) | IL Dosage (g) | TRS Yield (wt.%) |
|-----|-----|-----|-----|
| 1 | 100 | 0.5 | 16.65 |
| 2 | 0 | 0.5 | 9.50 |
| 3 | 100 | 0 | 6.16 |
| 4 | 0 | 0 | 1.78 |

In summary, efficient mass and heat transfer rely on free and sufficient molecular motion, which enhances the ability of solvent and catalyst molecules to diffuse and penetrate into the interior of sugarcane bagasse cellulose, making it easier for H$^+$ to approach the substrate and participate in hydrolysis reactions. Excessive viscosity is not conducive to molecular movement, and during hydrolysis, there may be a local high temperature leading to carbonization, so it should be avoided. The presence of a small amount of water can promote ionization and the production of more protons, thus facilitating hydrolysis. Under the above optimal conditions, the maximum yield of 33.97% for TRS was obtained at last.

### 3.1.4. Comparison on Pretreatment with Different ILs

In order to compare the effects of other ionic liquid pretreatments on the hydrolysis of sugarcane cellulose to prepare TRS, this study investigated the effects of [Amim]Cl, [Bmim]Br, [Emim]Br, and [Prmim]Br on the yield of hydrolysis to prepare TRS after pretreatment of sugarcane cellulose. The results are shown in Table 2. In order to eliminate measurement errors caused by the ionic liquid itself, the experimental group without adding sugarcane cellulose but with all other reaction conditions being consistent was used as the blank group, and the solution of the blank group was used as the baseline calibration for analyzing TRS. These four ionic liquids have no significant pretreatment effect on sugarcane cellulose compared to [Bmim]Cl, which has been introduced in Section 3.1. In addition, as the alkyl chain on the imidazole cation of the ionic liquid increases, the overall pretreatment effect on sugarcane cellulose gradually decreases. In the experiment, it can be observed that during the pretreatment process of sugarcane cellulose with the above three bromide ionic liquids, sugarcane cellulose almost does not dissolve, so it has little effect on subsequent hydrolysis. After pretreatment with [Bmim]Cl, sugarcane cellulose is completely dissolved in the hydrolysis system, reducing the polymerization degree and particle size of the raw material, thereby accelerating the reaction. The hydrogen bonding network of sugarcane cellulose itself is also disrupted, exposing it to H$^+$ attack, which is very beneficial for hydrolysis.

**Table 2.** The comparison of the different ILs used in pretreatment about the TRS yield.

| ILs | TRS Yield (wt.%) | | |
| --- | --- | --- | --- |
| | **1 h** | **2 h** | **3 h** |
| [Amim]Cl | 7.99 | 5.05 | 2.87 |
| [Emim]Br | 4.78 | 1.55 | 1.42 |
| [Prmim]Br | 3.96 | 1.12 | 1.27 |
| [Bmim]Br | 3.09 | 1.91 | 0.80 |

### 3.2. Characterization on Products after Hydrolysis

In order to compare the effect of water on the structure and other properties of sugarcane cellulose after hydrolysis at the pretreated temperature (i.e., 100 °C), FT-IR, SEM, and TGA technologies were used to characterize the hydrolyzed sugarcane cellulose, and their possible structure, surface morphology, and thermal properties were compared, as shown in Figures 4–6. For the involved test samples, sample a is the sugarcane cellulose sample after pretreatment with [Bmim]Cl; the system does not add additional water and is directly hydrolyzed at 100 °C; sample b is the sugarcane cellulose sample after pretreatment with [Bmim]Cl; the system added a small amount of additional water and then hydrolyzed it at 100 °C; and sample c is the sugarcane cellulose sample without [Bmim]Cl pretreatment, adding a considerable amount of water directly to the system and then hydrolyzing.

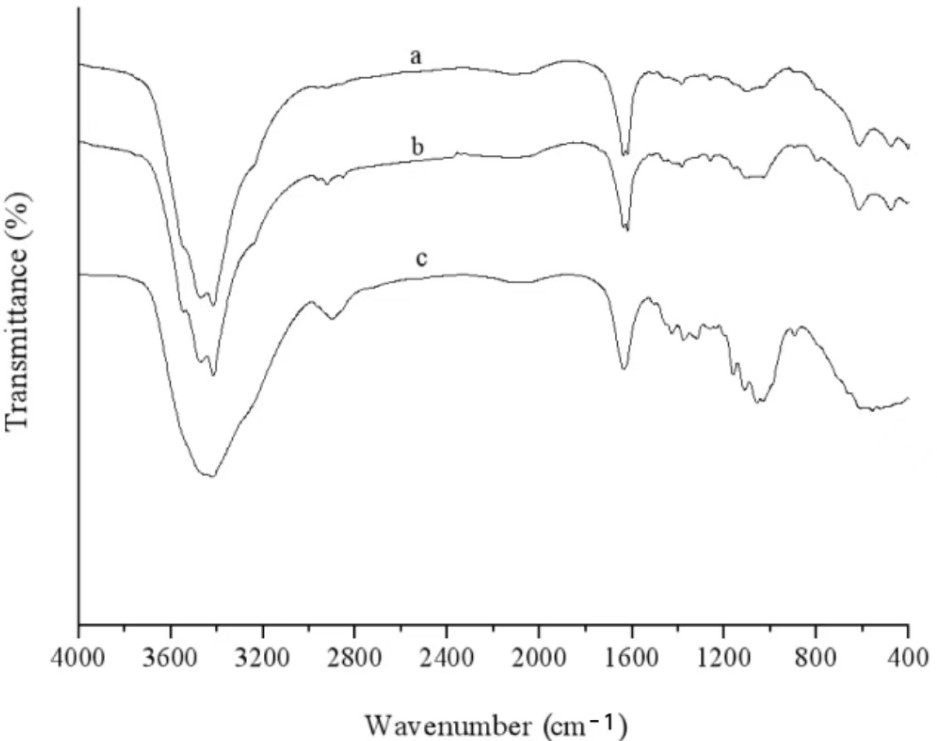

**Figure 5.** FT-IR spectra of cellulose after hydrolysis (sample a was pretreated with [Bmim]Cl and no water was added for hydrolysis; sample b was pretreated with [Bmim]Cl and water was added for hydrolysis; sample c was not pretreated with [Bmim]Cl and water was added for hydrolysis).

Figure 5 shows the FT-IR spectra of the powder of the above three sugarcane cellulose samples. It can be seen that the spectrum c is almost consistent with that of the original sugarcane cellulose sample, indicating that the untreated sugarcane cellulose does not undergo an obvious chemical reaction in pure water. In the spectrum of sample c, the peak at 1428 cm$^{-1}$ is attributed to the shearing vibration of -CH$_2$ connected to O-6 of cellulose chains. Compared with the spectrum of sample c, the intensity of this peak in the spectra of samples a and b is significantly weakened, indicating that the intramolecular hydrogen

bond with sugarcane cellulose O-6 is destroyed [21]. The peaks at 1162~1058 cm$^{-1}$ are attributed to the C-O-C stretching vibration of sugarcane cellulose, and the absorption intensity of this peak in the spectra of samples a and b is also weakened, indicating that C-O-C in sugarcane cellulose after hydrolysis of these two systems has been destroyed to a certain extent. Compared with the spectrum of sample c, the O-H vibration peaks (1600~1650 cm$^{-1}$) and peaks of sugarcane cellulose in the spectra of samples a and b indicate that the hydrogen bonds of sugarcane cellulose after hydrolysis in these two systems are opened. However, the spectra of sugarcane cellulose samples a and b are very similar, and their structures are basically the same, indicating that after pretreatment by [Bmim]Cl, a small amount of additional water is added to the system to precipitate part of sugarcane cellulose from the ionic liquid homogeneous system, and there is no significant difference in the FT-IR spectrum after hydrolysis, which may be related to the ionic liquid [Bmim]Cl itself carrying a small amount of binding water. The above results are similar to those in previous studies [41,42].

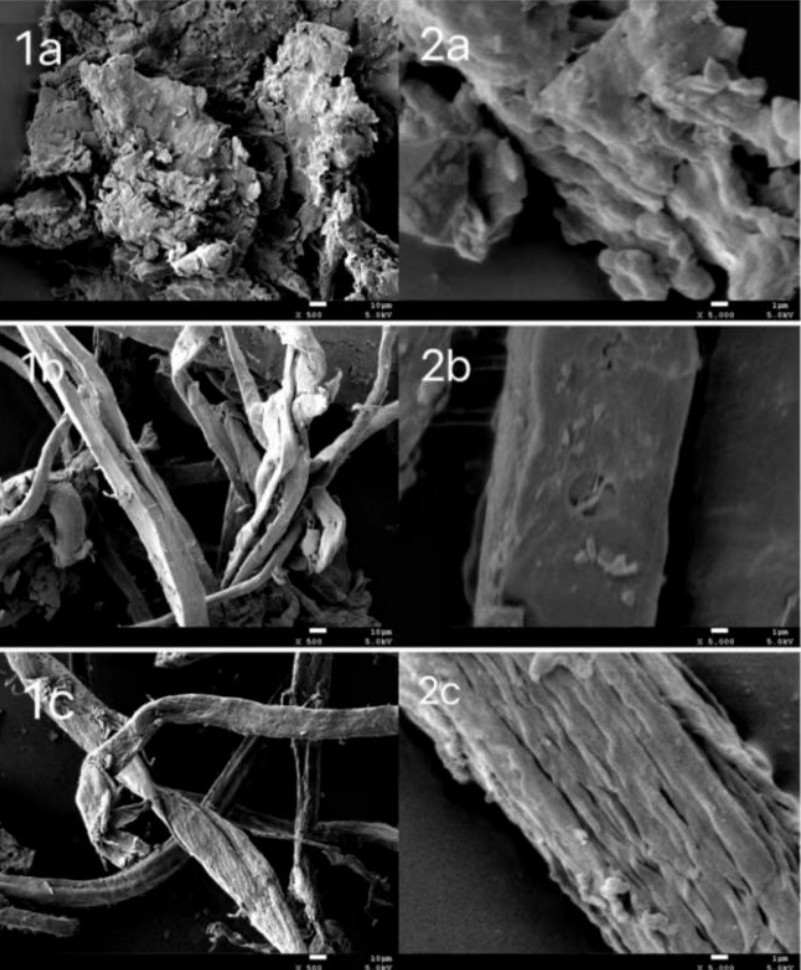

**Figure 6.** SEM images of cellulose at different hydrolysis conditions (**1a**–**1c**): ×500; (**2a**–**2c**): ×5000; sample a was pretreated with [Bmim]Cl and no water was added for hydrolysis; sample b was pretreated with [Bmim]Cl and water was added for hydrolysis; sample c was not pretreated with [Bmim]Cl and water was added for hydrolysis).

Figure 6 shows the SEM images of the powder of the above three sugarcane cellulose samples. It can be intuitively seen that the surface morphology of the three sugarcane cellulose samples is different. The surface of sample a is rough, and the original fiber structure of sugarcane cellulose is almost destroyed, forming a relatively loose, massive structure. The observation ratio of the electron microscope continued to enlarge, and it was

found that it has a fine porous structure and that the surface particle size is not uniform. In the SEM observation, it can be found that sample b shows a small number of SEM images similar to sample A, that is, the damaged massive structure, but most of the structure, as shown in Figure 6(1b,2b), a relatively complete fiber structure is maintained, and the surface is relatively smooth, which may be because a small amount of extra water is added after the sugarcane cellulose is pretreated, and the water added in this process makes it partially sweet. The precipitation of sugarcane cellulose in the ionic liquid system resulted in different degrees of hydrolysis of sugarcane cellulose, resulting in two different structures. Sample c was directly pretreated with water instead of IL. In the SEM observation, it can be found that all sugarcane cellulose maintained the original fiber structure and was not damaged, but its surface is no longer smooth.

Figure 7 shows the TG curves of the above three sugarcane cellulose powder samples. It can be seen that the TG curves of sample a and sample c almost overlap, and the initial decomposition temperature of sample a (285 °C) is slightly lower than that of sample c (301 °C), which is 354 °C and 349 °C, respectively. To reach the maximum decomposition rate, the entire weight loss process mainly occurs between 285~377 °C and 301~381 °C, when the initial decomposition temperature of sample b is close to that of sample c and the weight loss rate reaches the maximum value at 352 °C. It can be intuitively seen from the figure that the temperature of sample b reaches 800 °C. The residual mass of sample a and sample c is only 23.26% and 22.42%, respectively. These results are similar to those in current reports [43], indicating that the thermal stability of sugarcane cellulose is slightly reduced after pretreatment with ionic liquid [Bmim]Cl.

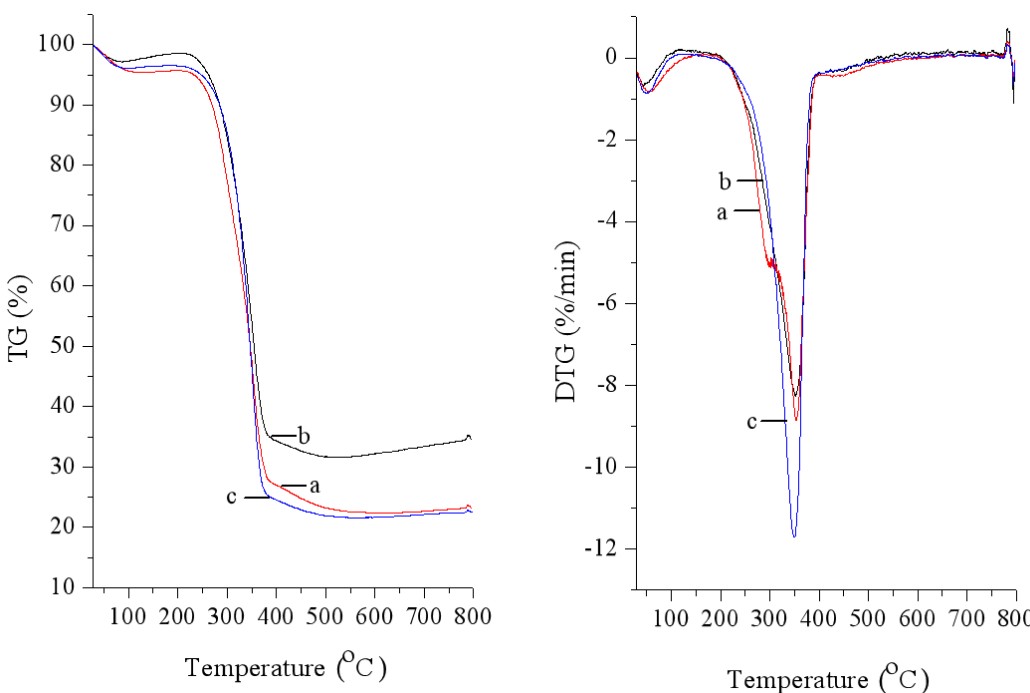

**Figure 7.** TG and DTG curve of cellulose at different hydrolysis conditions. (red line: sample a, black line: sample b, and blue line: sample c).

*3.3. Heterogeneous Catalytic Hydrolysis with [C₃SO₃Hmim]HSO₄*

Unlike the previous homogeneous hydrolysis, in the following heterogeneous hydrolysis experiment, sugarcane cellulose was not pretreated. Sugarcane cellulose exists in solid form in the system, while catalyst ionic liquids exist in the liquid phase. In heterogeneous systems, to a certain extent, water serves as the solvent, and acidic ionic liquids serve as catalysts. This is a multiphase reaction system, and sugarcane cellulose exists in the form of solid particles in the system. At the beginning, the ionic liquid diffuses onto the surface of sugarcane cellulose particles and undergoes hydrolysis. After the sugarcane cellulose on

the surface is hydrolyzed, the internal hydrolysis reaction can continue. Temperature has a significant impact on the hydrolysis process of cellulose. At 80 °C, the yield of TRS is very low. As the temperature increases, the viscosity of the IL solution system decreases while also enhancing the ionization of $H_2O$, thereby accelerating the hydrolysis of sugarcane cellulose to obtain TRS. However, as the temperature continues to rise, TRS will slowly degrade into some small molecule compounds, such as 5-carboxymethyl furfural (5-HMF), resulting in a decrease in yield. Overall, higher temperatures may not necessarily lead to an increase in TRS yield, but they can cause the maximum value of TRS to appear earlier. It can be seen from the results of gas chromatographic (GC) analysis of the extract of the aqueous phase that there are indeed small molecules such as 5-HMF present, as shown in Figure 8 of gas chromatograms with the standard sample. The peak at 8.2 min corresponds to 5-HMF, and its yield also increases with the increase in temperature. Under the continuous effects of acidic catalysts and high temperatures, cellulose is first hydrolyzed to obtain glucose. Secondly, the oxygen on the glucuraldehyde group forms intramolecular hydrogen bonds with the adjacent hydroxyl group, forming a five-membered ring. Then the protons transfer to form fructose, and finally, three water molecules are removed from fructose to produce 5-HMF. Therefore, when the reaction time is too long, the yield of TRS will decrease while the yield of 5-HMF will increase, which accords with the results of the previous study [44]. In addition, it can be clearly observed from the system appearance during the experiment that the reaction solution gradually begins to change color with the increase in temperature, from light yellow to brownish yellow, and even eventually turns brown. In addition, the investigation of hydrolysis time showed that the TRS content produced by sugarcane cellulose hydrolysis at 100 °C increased continuously with the extension of time, reaching its maximum value at 5 h. Then, with the extension of time, the TRS content began to decrease again, and more and more TRS degraded, resulting in a decrease in yield. Through GC analysis, it can be seen that with the extension of time, the content of 5-HMF at 140 °C also increases. After optimization on the IL concentration, reaction temperature, and time, the highest TRS yield of 5.98% was finally achieved under the conditions of 4 mol/L [C₃SO₃Hmim]HSO₄, 140 °C, and 1 h, which was lower than the level of the aforementioned homogeneous hydrolysis.

*3.4. Separation of Ionic Liquids from Reducing Sugars*

In this section, strongly acidic cation exchange resin 732H with −SO₃H group was used to separate and purify the ionic liquid in the sugarcane cellulose hydrolysis system and the mixed solution of TRS. The resin adsorbs and exchanges the cation of the ionic liquid. After adsorbing ionic liquid, the resin was eluted with hydrochloric acid, and then the eluent was concentrated under vacuum to obtain the ionic liquid.

In the experiment, two homogeneous systems were taken as the object, namely the solution system formed by the catalytic hydrolysis of sugarcane cellulose by [C₃SO₃Hmim]HSO₄ and [HBth][CH₃SO₃], respectively, to study the separation efficiency of ionic liquid and reducing sugar. The results are shown in Figure 9a,b. It should be noticed in Figure 9b that the same amount of [Bmim]Cl was added to the hydrolysis solution of [C₃SO₃Hmim]HSO₄ in order to observe the selective separation results of the resin for the two ILs according to Figure 9a.

It can be seen that 732H resin shows excellent exchange-adsorption capacity for the ionic liquid [Bmim]Cl, and the adsorption rate of [Bmim]Cl can reach about 96%. However, for catalyst ionic liquids [C₃SO₃Hmim]HSO₄ and [HBth][CH₃SO₃], 732H resin shows significantly different adsorption effects. The resin shows a very good adsorption performance on [HBth][CH₃SO₃] (100%), but almost no adsorption effect on [C₃SO₃Hmim]HSO₄. The reason may be that the cations in [C₃SO₃Hmim]HSO₄ have a sulfonic acid group, and it is difficult to replace the sulfonic acid group with hydrogen on the resin. In addition, due to the presence of a benzene ring, [HBth][CH₃SO₃] has a relatively strong π-π interaction with the resin, which enhances the binding force between the cation part of the ionic liquid and the resin, and the adsorption rate of the resin to TRS is also relatively small, which is also expected. It can be seen from the data in the figure that in the [HBth][CH₃SO₃]

catalytic system, the ionic liquid can be well separated from the product TRS, and the separation effect is very good. However, in the $[C_3SO_3Hmim]HSO_4$ catalytic system, only the solvent [Bmim]Cl can be effectively separated, but the catalyst $[C_3SO_3Hmim]HSO_4$ and the product TRS cannot be separated.

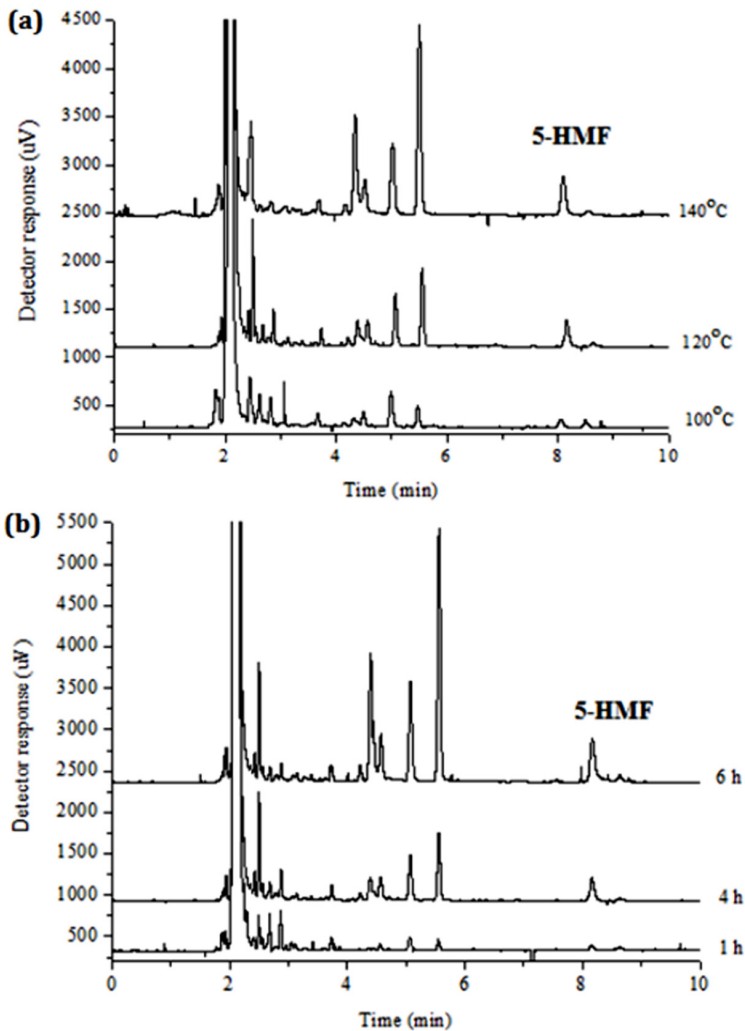

**Figure 8.** GC analysis results for the products under (**a**) different temperatures (100–120 °C) and (**b**) durations (1–6 h).

Resin is statically adsorbed with an ionic liquid and immersed in hydrochloric acid for desorption. The influence of different HCl concentrations as eluent on the desorption rate of an ionic liquid was investigated, and the results are shown in Table 3 for resin and $[Bmim]^+·[HBth]^+$-loaded resin. The results show that $[Bmim]^+$-loaded resin and $[Bmim]^+·[HBth]^+$-loaded resin are, respectively, prepared with 25 mL HCl solution with a mass concentration of 15%. After resin removal, the desorption rates of [Bmim]Cl reached 92.05% and 90.04%, respectively. However, the desorption rate of the $[HBth][CH_3SO_3]$ cation is very low, only 52.95%, which may be caused by the existence of a relatively strong π-π interaction between the cation and the resin, and it is difficult to desorp.

Finally, the recovered $[HBth][CH_3SO_3]$ was put into the next hydrolysis process to evaluate its repeatability, and the pretreatment and hydrolysis conditions in recycling are the same as before. The reused performance was still measured using the yield of TRS, with the first round yield considered to be 100%. Through the results in Figure 10, it can be found that after repeated recovery and use five times, the performance of the IL begins to show a significant decline, indicating that impurities contained in them have accumulated to a

certain extent, which will significantly interfere with the catalytic effect of the IL. At this time, multiple resin purifications are required to ensure sufficient purity of [HBth][CH$_3$SO$_3$].

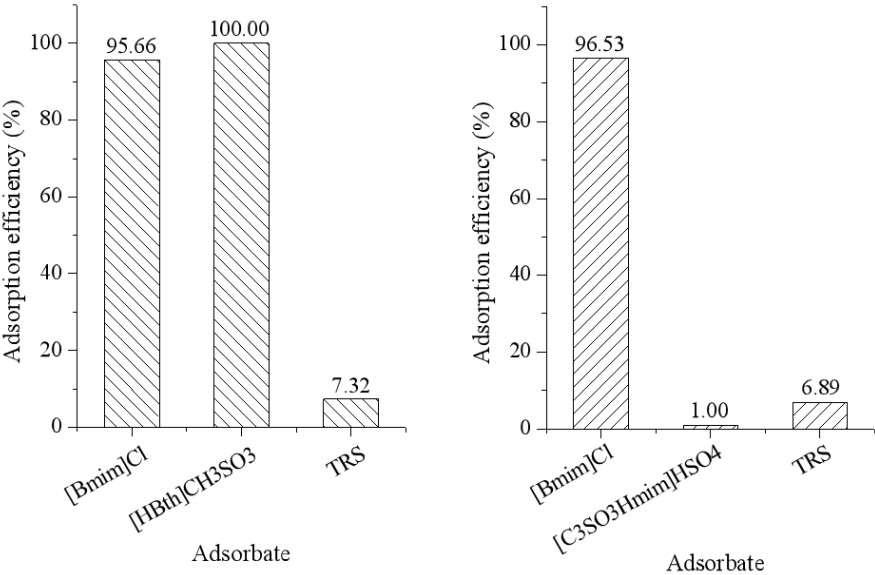

**Figure 9.** The comparison of adsorption capacity in two kinds of cellulosic hydrolysate catalyzed by [HBth][CH$_3$SO$_3$] and [C$_3$SO$_3$Hmim]HSO$_4$ with the coexistance of [Bmim]Cl.

**Table 3.** The effect of HCl concentration on desorption efficiency (732H resin).

| HCl Concentration (%) | [Bmim]$^+$-Loaded Resin | [Bmim]$^+$·[HBth]$^+$-Loaded Resin | |
|---|---|---|---|
| | [Bmim]Cl Desorption Rate (%) | [Bmim]Cl Desorption Rate (%) | [HBth]$^+$ Desorption Rate (%) |
| 1 | 16.55 | 12.47 | 5.58 |
| 5 | 58.01 | 55.10 | 28.35 |
| 10 | 77.06 | 73.95 | 48.55 |
| 15 | 92.05 | 90.04 | 52.95 |

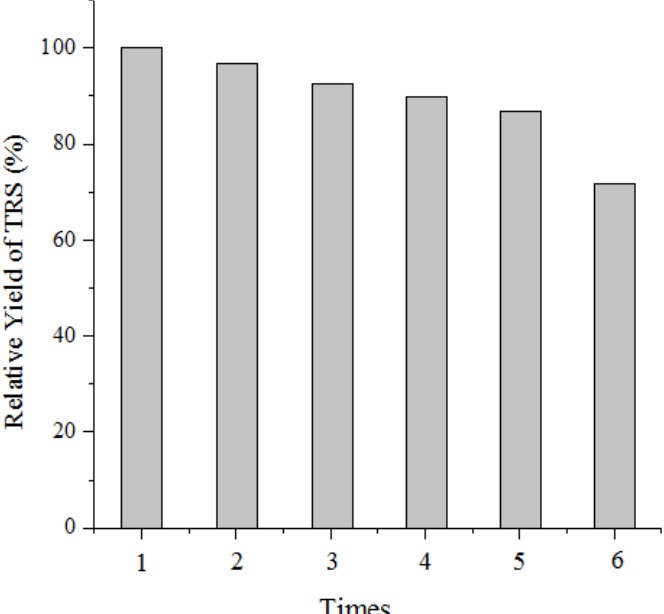

**Figure 10.** The reused performance of IL in recycling of six times.

## 4. Conclusions

In this study, the basic work of hydrolysis of sugarcane cellulose in a pure ionic liquid system of [HBth][CH$_3$SO$_3$] and [C$_3$SO$_3$Hmim]HSO$_4$ in two comparative modes without any other catalyst was studied for the first time. The main conclusions are as follows: The hydrolysis effect of ionic liquids is inevitably closely related to their structures, and their anions and cations must have strong interactions with intramolecular chains of cellulose. In addition, sufficient acidity must also be ensured. Those anions and cations containing more active hydrogen can cause an increase in the proton concentration in the hydrolysis system, leading to the destruction of hydrogen bonds in the amorphous region of cellulose, and then the hydrolysis of cellulose in the amorphous region generates water-soluble sugars. The pretreatment, IL concentration (dosage), reaction temperature, and time were all discovered as key conditions for hydrolysis. A higher temperature does not necessarily lead to an increased TRS yield, but it will make the maximum TRS appear earlier. At 100 °C, the TRS content produced by sugarcane cellulose hydrolysis increased with time until the maximum at 5 h, after which the TRS content began to decrease again with time. The hydrolysis pattern at 120 °C was similar to that at 100 °C, but the hydrolysis efficiency was slightly higher than that at 100 °C. However, at 140 °C, TRS decreased earlier. At 180 °C, the TRS yield curve became steeper, and the maximum value appeared after only 1 h and then began to decline rapidly.

The acidity of an acidic ionic liquid is an important factor affecting the hydrolysis of sugarcane cellulose. In the homogeneous hydrolysis system, the pretreatment ionic liquid [Bmim]Cl was used as the solvent to a certain extent, and the acidic ionic liquid [HBth][CH$_3$SO$_3$] was used as the catalyst. After pretreatment, the sugarcane cellulose was completely or partially dissolved in the ionic liquid, and the sugarcane cellulose molecules were dispersed to a great extent. Its hydrogen bond network is interrupted to a certain extent, exposing it to the attack of H$^+$, and the hydrolysis efficiency is higher than that of the heterogeneous system. In a homogeneous hydrolysis system, the hydrolysis of sugarcane cellulose to TRS by other ionic liquids ([Amim]Cl, [Bmim]Br, [Emim]Br, and [Prmim]Br) was compared. The results showed that the pretreatment effect of these four ionic liquids on sugarcane cellulose was not as good as [Bmim]Cl. Moreover, the effect of ionic liquid with Cl$^-$ on the pretreatment of sugarcane cellulose is more obvious than that of ionic liquid with Br$^-$. In the homogeneous hydrolysis system, the catalytic effect of the ionic liquid benzothiazole-methane sulfonate on sugarcane cellulose was also investigated. Moreover, the products were compared in a series of ways; among them, FT-IR characterization results showed that there was no new peak in sugarcane cellulose after hydrolysis, and the infrared spectrum was basically consistent with the characteristic peak of sugarcane cellulose before the reaction, indicating that there was no derivatization reaction between sugarcane cellulose and ionic liquid during the hydrolysis process. The SEM results showed that the surface of sugarcane cellulose before hydrolysis was relatively complete, smooth, and orderly, and the structure was obviously fibrous. After hydrolysis, the complete and orderly fiber structure of sugarcane cellulose was destroyed, and the surface became rough and loose. The TGA results showed that the thermal stability of sugarcane cellulose decreased after hydrolysis. In the heterogeneous hydrolysis system, water can be regarded as the solvent and the acidic ionic liquid [C$_3$SO$_3$Hmim]HSO$_4$ as the catalyst, to a certain extent. Sugarcane cellulose exists in the system in the form of solid particles, and H$^+$ can only attack the outer surface of sugarcane cellulose particles for hydrolysis. In the above two hydrolysis modes, the involved ionic liquids could be recovered and recycled, and the related resin post-treatment procedures are expected to reduce the cost during their large-scale use, thereby improving the economic benefits of the hydrolysis process.

**Supplementary Materials:** The following supporting information can be downloaded at: https://www.mdpi.com/article/10.3390/biomass4030049/s1, Figure S1: The standard curve of glucose; Figure S2: The standard curve of [Bmim]Cl; Figure S3: The standard curve of [C$_3$SO$_3$Hmim]HSO$_4$; Figure S4: The standard curve of [[HBth][CH$_3$SO$_3$].

**Author Contributions:** Conceptualization, R.L. and J.L.; methodology, R.L.; software, J.L.; validation, R.L. and E.L.; formal analysis, E.L.; investigation, E.L.; data curation, Z.L.; writing—original draft preparation, R.L.; writing—review and editing, A.A.; visualization, Z.L.; supervision, Z.L.; project administration, Z.L.; funding acquisition, S.Y. All authors have read and agreed to the published version of the manuscript.

**Funding:** This work is supported by Chengdu Key Research and Development Supporting Project (No. 2022-YF05-00910-SN).

**Institutional Review Board Statement:** Not applicable.

**Informed Consent Statement:** Not applicable.

**Data Availability Statement:** Data is contained within the article or Supplementary Materials.

**Acknowledgments:** All the authors' affiliations provided the convenience for related studies, respectively. We would like to thank the support from Center of Engineering Experimental Teaching, School of Chemical Engineering, Sichuan University, SEM image by Yanping Huang and FT-IR (Perkin-Elmer, Waltham, MA, USA) characterization by Lin Xiang.

**Conflicts of Interest:** The authors declare no conflicts of interest.

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
