# Peer review of "Ionic Liquid Catalyzed Hydrolysis of Sugarcane Cellulose to Produce Reducing Sugar"

_2673-8783, doi:10.3390/biomass4030049_

Round 1
Reviewer 1 Report
Comments and Suggestions for Authors
The manuscript describes the hydrolysis of sugarcane cellulose in the presence of different ionic liquids. The topic of cellulose hydrolysis in the ionic liquids is quite explored, while the authors do not cite most of the state of the art and do not relate to previous studies (e.g. RSC Adv., 2018,8, 14623-14632, Polysaccharides 2022, 3(4), 671-691, Fuel, 2016, 164, 46-50, ACS Sustainable Chem. Eng. 2016, 4, 6, 3352–3356). Moreover, it is not clear why the authors choose the 2 ionic liquids they investigate- especially the one based benzothiazole is not very common (there are different more common protic ILs).
The manuscript lacks consistency in many places and is imprecise.
More considerations are provided below:
The authors claim that, “the large hydrogen bond in cellulose affects its hydrolysis effect” – do the authors mean the large number of H-bonds?
The authors write that advantages and disadvantages of cellulose hydrolysis methods “are described above”, whereas there is no discussions on this in the introduction. Do the authors mean the citing literature?
What do authors mean by pretreatment “with 100 times”?
The materials and methods section is very long. I suggest to transfer part of it into the SI (e.g. the standard curves).
Could the authors explain why the hydrolysis yield decreased with higher loading of the ionic liquid?
The description of Figure 6 does not provide information what a, b and stands for. The same applies to figure 7.
The desorption of [HBth][CH3SO3] with HCl is unclear. Moreover, the information how the recycling studies were conducted is missing. The authors describe that the protic IL was added into the IL that was used for the pretreatment ([Bmim]Cl,). What was used in the recycling experiment and did it include the pretreatment process? (from the desorption process it seems mostly [Bmim]Cl was recovered).
The authors claim that hydrolysis of cellulose using sulfuric acid or hydrochloric acid are not environmentally and personnel friendly. Could the authors relate to the cost of the catalyst they explored which provided only ca. 25% yield of TRS relative to 72% obtained when sulfuric acid was used or 42% when hydrochloric acid was used?
Dots at the end of many sentences are missing. The other have double dots.
Comments on the Quality of English LanguageEnglish Language needs to be improved.
Author Response
Reviewer 1
The manuscript describes the hydrolysis of sugarcane cellulose in the presence of different ionic liquids. The topic of cellulose hydrolysis in the ionic liquids is quite explored, while the authors do not cite most of the state of the art and do not relate to previous studies (e.g. RSC Adv., 2018,8, 14623-14632, Polysaccharides 2022, 3(4), 671-691, Fuel, 2016, 164, 46-50, ACS Sustainable Chem. Eng. 2016, 4, 6, 3352–3356). Moreover, it is not clear why the authors choose the 2 ionic liquids they investigate- especially the one based benzothiazole is not very common (there are different more common protic ILs).
Response: Thanks for your reviewing! We know that there are still many shortcomings in our work and content, and it is hoped that they can be improved through this revision. Thank you for your support and consideration! During revision, all of above references have been cited in our introduction during revision, and the ILs in this study were screened through our pilot researches. Currently, most ILs used in biomass field belong to imidazolium type, so more ILs with other types need to be paid enough attention on; meanwhile the comparison on different hydrolysis modes is meaningful. Considering benzothiazolium ILs have been applied as new effective Brønsted-acidic catalyst [32] and imidazolium IL-grafted heterogeneous catalyst was proved success in cellulose hydrolysis [33], it is worth exploring if the former is capable of handling the task of hydrolyzing sugarcane cellulose, and the latter can be directly employed in heterogeneous catalysis.
The manuscript lacks consistency in many places and is imprecise.
More considerations are provided below:
The authors claim that, “the large hydrogen bond in cellulose affects its hydrolysis effect” – do the authors mean the large number of H-bonds?
Response: Yes, and the expression has been revised.
The authors write that advantages and disadvantages of cellulose hydrolysis methods “are described above”, whereas there is no discussions on this in the introduction. Do the authors mean the citing literature?
Response: Sorry for the unclear expression, and the sentence has been replaced by “They all have their own inherent advantages and disadvantages, and exploring mild, green, and efficient degradation technologies has always been a common goal in both academia and industry”.
What do authors mean by pretreatment “with 100 times”?
Response: Sorry for the error and “times” has been changed with “℃”.
The materials and methods section is very long. I suggest to transfer part of it into the SI (e.g. the standard curves).
Response: Thanks for your guidance! The figures of standard curves and related details have been removed from the text to SI in order to make the content more concise.
Could the authors explain why the hydrolysis yield decreased with higher loading of the ionic liquid?
Response: Thanks for your guidance! As you may know, when the concentration of ionic liquids increases, the viscosity of the entire system increases, affecting its permeation in cellulose as well as heat and mass transfer. If the concentration is too low, it also affects the hydrolysis efficiency. Therefore, a moderate IL concentration is recommended. Above analysis has been added in section 3.1.1.
The description of Figure 6 does not provide information what a, b and stands for. The same applies to figure 7.
Response: Thanks for your guidance! The necessary description has been added in the caption of the figures.
The desorption of [HBth][CH3SO3] with HCl is unclear. Moreover, the information how the recycling studies were conducted is missing. The authors describe that the protic IL was added into the IL that was used for the pretreatment ([Bmim]Cl,). What was used in the recycling experiment and did it include the pretreatment process? (from the desorption process it seems mostly [Bmim]Cl was recovered).
Response: Thanks for guidance! As described in Section 3.4, the recovered [HBth][CH3SO3] was put into next hydrolysis process to evaluate its repeatability, and the reused performance was still measured using the yield of TRS, with the first round yield considered to be 100%. The desorption results of [HBth][CH3SO3] with HCl were included in Table 3. The ionic liquids used for pretreatment and catalysis were both recovered in desorption, and the pretreatment and hydrolysis conditions in recycling are the same as before.
The authors claim that hydrolysis of cellulose using sulfuric acid or hydrochloric acid are not environmentally and personnel friendly. Could the authors relate to the cost of the catalyst they explored which provided only ca. 25% yield of TRS relative to 72% obtained when sulfuric acid was used or 42% when hydrochloric acid was used?
Response: Thanks for guidance! 72% and 42% refer to the concentration of two acids rather than the TRS yield they yield. The cost of ionic liquids is truly higher than that of traditional inorganic acids with the same volume, but they can be recycled as green solvents, making them more environmentally friendly, safe, and healthy for operators. Meanwhile the hydrolysis time is shorter. There are also many successful industrial cases now, proving the economic feasibility of their actual application.
Dots at the end of many sentences are missing. The other have double dots.
Response: Thanks for guidance! The problem has been checked and revised.
Reviewer 2 Report
Comments and Suggestions for Authors
TITLE
Acceptable
ABSTRACT
- Provide more specific details on the experimental approach, including the specific ILs used, experimental conditions (e.g., temperatures, reaction times), and key results.
- Please add specific quantitative results such as reaction yields or separation efficiencies.
INTRODUCTION
- Provide more specific details on the experimental approach, including the specific ILs used, experimental conditions (e.g., temperatures, reaction times), and key results.
- Provide additional Background on ILs and their unique properties, such as low vapor pressure, tunable solvation behavior, and wide liquid range.
- Add how the investigation of cellulose hydrolysis within pure IL systems fills a critical gap in current knowledge and study the intrinsic catalytic properties of ILs.
MATERIALS AND METHODS
- Please provide specific details for each reagent, such as concentration, purity, and source (supplier).
- Explain why choosing that cellulose-to-IL ratios, reaction temperatures, and times.
- Specify the exact quantities of [C3SO3Hmim]HSO4 ionic liquid and water added, as well as the reaction temperature and duration.
- Explain the phenol-concentrated sulfuric acid method in more detail, including the specific steps involved in sample preparation, color development, and absorbance measurement.
- What is the dilution series used to generate the calibration curve?
- What is the dilution ratios and volumes used for each standard solution?
RESULTS AND DISCUSSION
- Specify the pre-treatment methods used with [Bmim]Cl in more detail. Explain how these methods reduce the degree of polymerization and increase surface area.
- Explain why the yield of hydrolyzed reducing sugar during pre-treatment is low (0.75%) and why this step is crucial for subsequent reactions.
- Why the yield initially increases with time at low catalyst dosages but decreases at higher dosages.
- Please discuss how molecular movement, viscosity changes, and ionization of water influence the rate of hydrolysis.
- Why the yield of reducing sugar decreases at higher temperatures due to degradation into small molecules like 5-carboxymethyl furfural.
- Compare the effects of various ionic liquids on the hydrolysis of sugarcane cellulose.
- How the structural changes observed in sugarcane cellulose samples after hydrolysis correlate with their pretreatment methods and hydrolysis conditions.
CONCLUSION
- Please mention if the study is the first to investigate the hydrolysis of sugarcane cellulose in a pure ionic liquid system without additional catalysts.
- Include the effectiveness of different ionic liquids in pretreatment and catalysis, the impact of homogeneous vs. heterogeneous hydrolysis systems, and the structural changes observed in sugarcane cellulose after hydrolysis.
Author Response
Reviewer 2
ABSTRACT
- Provide more specific details on the experimental approach, including the specific ILs used, experimental conditions (e.g., temperatures, reaction times), and key results.
Response: Thanks for guidance! The details have been revised according to your requirements.
- Please add specific quantitative results such as reaction yields or separation efficiencies.
Response: Thanks for guidance! The results have been revised according to your requirements.
INTRODUCTION
- Provide more specific details on the experimental approach, including the specific ILs used, experimental conditions (e.g., temperatures, reaction times), and key results.
Response: Thanks for guidance! The details have been revised according to your requirements.
- Provide additional Background on ILs and their unique properties, such as low vapor pressure, tunable solvation behavior, and wide liquid range.
Response: Thanks for guidance! These backgrounds have been added in the section of Introduction.
- Add how the investigation of cellulose hydrolysis within pure IL systems fills a critical gap in current knowledge and study the intrinsic catalytic properties of ILs.
Response: Thanks for guidance! Although there are many types of ionic liquids currently available, the use of other ILs rather than imidazolium ILs is relatively limited, and there is even less research on their performance comparison under different catalytic modes and conditions. Related hydrolysis results and difference are still unknown. Based on this, this research aimed to fill the relevant research gap and provide meaningful reference for researchers in this field.
MATERIALS AND METHODS
- Please provide specific details for each reagent, such as concentration, purity, and source (supplier).
Response: Thanks for guidance! Related information has been added.
- Explain why choosing that cellulose-to-IL ratios, reaction temperatures, and times.
Response: Thanks for guidance! They are the main parameters in current studies of cellulose hydrolysis after the suitable IL is selected. The level of cellulose-to-IL ratio, reaction temperature, and time determine the consumption of the reagents and energy as well as the efficiency and cost of the whole process. Therefore, like those similar research literature, we explored their influence patterns.
- Specify the exact quantities of [C3SO3Hmim]HSO4 ionic liquid and water added, as well as the reaction temperature and duration.
Response: Their ranges were 0.1~1.0 g, 50~500μL, 100~180℃ and 1~3h, respectively, and the optimal conditions in these ranges were explored in the following studies. These information has been added in the section 2.3
- Explain the phenol-concentrated sulfuric acid method in more detail, including the specific steps involved in sample preparation, color development, and absorbance measurement.
Response: The details have been described in Section 2.4. That is, 0.1 mL of the supernatant was taken to be measured and diluted to a certain multiple, then a pipet was used to measure a certain volume of the diluted liquid to be measured in a 25 mL colorimetric tube, and was mixed with water to 2 mL. Only 2 mL of water is added into one of the colorimetric tubes as the reference solution. 1 mL of freshly prepared 5% phenol solution was added and shaken well, then 5 mL of concentrated H2SO4 was added and shaken well; after cooled at room temperature for 20 min, the color of the system turn to orange and the absorbance at 490 nm was determined.
- What is the dilution series used to generate the calibration curve?
Response: Thanks for guidance! The series include the followings:
For 40 μg/mL glucose mother liquor, 0.1 mL, 0.2 mL, 0.3 mL, 0.4 mL, 0.5 mL, 0.7 mL, or 0.8 mL of this mother liquor was taken and mixed with UP water to 2.0 mL for developing calibration curve.
For 200 μg/mL[C3SO3Hmim]HSO4 mother liquor, 0.1 mL, 0.2 mL, 0.3 mL, 0.3 mL, 0.4 mL, 0.5 mL, 0.6 mL, 0.7 mL,or 0.8 mL of this mother liquor was diluted with UP water to 10 mL for developing calibration curve.
For 200 μg/mL [Bmim]Cl mother liquor, 0.2 mL, 0.4 mL, 0.6mL, 0.8 mL, 1.0 mL, 1.2 mL 1.4 mL 1.6 mL, or 2.0 mL of this mother liquor was diluted with UP water to 10 mL for developing calibration curve.
For 200 μg/mL[HBth][CH3SO3] mother liquor, 0.1 mL, 0.2 mL, 0.4mL, 0.5 mL, 0.6 mL, 0.8 mL, 1.0 mL, 1.2 mL or 1.3 mL of this mother liquor was diluted with UP water to 10 mL for developing calibration curve.
- What is the dilution ratios and volumes used for each standard solution?
Response: Thanks for guidance! According to above response to the previous comment, the volume was 2 or 10 mL, and dilution ratios were 1:20~8:20, 1:100~8:100, 1:50~1:5 and 1:100~13:100, respectively.
RESULTS AND DISCUSSION
- Specify the pre-treatment methods used with [Bmim]Cl in more detail. Explain how these methods reduce the degree of polymerization and increase surface area.
Response: Sorry for our negligence! The details of pre-treatment process with [Bmim]Cl has been introduced in Section 2.3.1. 50 mg of sugarcane cellulose was added into a flask containing 2 g [Bmim]Cl, and the pretreatment was carried out at 100℃by stirring for 30 min. According to current reports of ILs applied in the cellulose field [40], the free Cl- and [Bmim]+ions in the ionic liquid interact with H and O atoms in cellulose hydroxyl groups, respectively. Due to the strong electronegativity of Cl-, its strong traction effect towards H atoms greatly weakens both the intermolecular and intramolecular hydrogen bonds within cellulose structures. When the charge of hydroxyl groups is dispersed to a high degree, the aggregated networks of cellulose will be disrupted, and then the molecular chains will be broken. These contents have been added in Section 3.1.
- Explain why the yield of hydrolyzed reducing sugar during pre-treatment is low (0.75%) and why this step is crucial for subsequent reactions.
Response: Thanks for guidance! Based on our responses to the previous comment, during the pretreatment process, cellulose mainly undergoes depolymerization of its network structure, and the cross-linking among different molecular chains will be broken, which can make it easier to generate TRS and achieve higher yields in the subsequent hydrolysis process. At the same time, due to the mild pretreatment conditions, it is not easy for cellulose to generate TRS in this stage, so its yield is only 0.75%.
- Why the yield initially increases with time at low catalyst dosages but decreases at higher dosages.
Response: Thanks for your guidance! As you may know, when the concentration of ionic liquids increases, the viscosity of the entire system increases, affecting its permeation in cellulose as well as heat and mass transfer. If the concentration is too low, it also affects the hydrolysis efficiency. Therefore, a moderate IL concentration is recommended. Above analysis has been added in section 3.1.1.
- Please discuss how molecular movement, viscosity changes, and ionization of water influence the rate of hydrolysis.
Response: Thanks for your guidance! Efficient mass and heat transfer rely on free and sufficient molecular motion, which enhances the ability of solvent and catalyst molecules to diffuse and penetrate into the interior of sugarcane bagassecellulose, making it easier for H+to approach the substrate and participate in hydrolysis reactions. Excessive viscosity is not conducive to molecular movement, and during hydrolysis, there may be local high temperature leading to carbonization, so it should be avoided. The presence of a small amount of water can promote ionization and the production of more protons, thus facilitating hydrolysis. These contents have been added in Section 3.1.3.
- Why the yield of reducing sugar decreases at higher temperatures due to degradation into small molecules like 5-carboxymethyl furfural (5-HMF).
Response: Thanks for your guidance! Under the continuous effects of acidic catalysts and high temperature, cellulose is first hydrolyzed to obtain glucose. Secondly, the oxygen on the glucuraldehyde group forms intramolecular hydrogen bonds with the adjacent hydroxyl group, forming a five-membered ring. Then the protons transfer to form fructose, and finally three water molecules are removed from fructose to produce HMF. Therefore, when the reaction time is too long, the yield of TRS will decrease while the yield of HMF will increase, which accords with the results in previous study [45]. These contents have been added in Section 3.3.
- Compare the effects of various ionic liquids on the hydrolysis of sugarcane cellulose.
Response: Thanks for your guidance! The hydrolysis effect of ionic liquids is inevitably closely related to their structures, and their anions and cations must have strong interactions with intramolecular chains of cellulose. In addition, sufficient acidity must also be ensured. Those anions and cations containing more active hydrogen can cause an increase in the proton concentration in the hydrolysis system, leading to the destruction of hydrogen bonds in the amorphous region of cellulose, and then the hydrolysis of cellulose in the amorphous region generates water-soluble sugars. In addition, the hydrolysis effect of homogeneous catalytic mode with IL is usually superior to heterogeneous catalytic mode. These contents have been added in Section 4.
- How the structural changes observed in sugarcane cellulose samples after hydrolysis correlate with their pretreatment methods and hydrolysis conditions.
Response: Thanks for your guidance! Based on the previous scheme and new Figure 2, the H-bonds will be broken in the network of cellulose during pretreatment, and the C-O in its chains will be broken during hydrolysis, which can be reflected through SEM images, TG and spectral analysis.
CONCLUSION
- Please mention if the study is the first to investigate the hydrolysis of sugarcane cellulose in a pure ionic liquid system without additional catalysts.
Response: Thanks for your guidance! In this study, the basic work of hydrolysis of sugarcane cellulose in pure ionic liquid system of [HBth][CH3SO3] and [C3SO3Hmim]HSO4 in two comparative modes without any other catalyst was studied for the first time.
- Include the effectiveness of different ionic liquids in pretreatment and catalysis, the impact of homogeneous vs. heterogeneous hydrolysis systems, and the structural changes observed in sugarcane cellulose after hydrolysis.
Response: Thanks for your guidance! The mentioned conclusions have been added in this section.
Reviewer 3 Report
Comments and Suggestions for Authors
Authors are recommended to carefully proofread the text. Punctuation marks and spaces, subscripts in formulae are missing in many sentences.
My review includes questions for the authors. These are not questions to satisfy my own curiosity. These are points in the text that were not clear to me as a reader. Accordingly, I assume that this may not be clear to other readers either. Therefore, if the question has a right to exist, I suggest correcting the text itself, instead of answering me in detail in a response without the text improvement, as some authors do. The purpose of a review is to make the manuscript understandable to the reader.
The experimental results obtained by the authors are very extensive and varied. Indeed, many types of pretreatments have been studied: with water, without water, homogeneous and heterogeneous reaction, different temperatures, etc, and this is an advantage of the study. On the other hand, it is complicated for a reader who was not directly involved in the experiment to navigate this “sea”. I strongly recommend making an experimental design figure, either as a graphic abstract or as a figure for the second or third part.
Unfortunately, in the third part, there are no references confirming the authors’ hypotheses.
I would recommend stylistically adjusting the abstract. Firstly, remove repetitions of phrases (if it is already stated in one sentence that you obtained reducing sugars from cellulose, there is no need to repeat this in the next one, etc.) and even entire sentences. Secondly, it is not necessary to specify which particular type of ion exchange resins you used; not all readers are familiar with this, it's better to mention it in the Materials and Methods section. Avoid using abbreviations, especially without clarification (TBS). In my opinion, it's better to use names rather than formulas of ionic liquids in the abstract. Since this is a research article, some resulting numbers in the abstract are encouraged.
Lines 34-35. The sentence need improvement. Large hydrogen bond? What is hydrolysis effect of cellulose? Is it an effect?
Line 37. I do not see description of advantages and disadvantages of these methods above. I am not saying that they are necessary - just the fact that they are not discussed.
Line 60. “there are few related reports”. The references are missed.
Check the text in line 65. Something is wrong with this sentence.
Line 69. “the base group of bagasse is divided into fibrin, hemicellulose and lignin”. What mean the base group? Also, check the information about component content of biomass.
The introduction lacks justification for why ionic liquids are suitable for bagasse hydrolysis. The authors have described the drawbacks of acidic and alkaline hydrolysis. Therefore, there is no choice but to resort to hydrolysis using ionic liquids? This was the motivation? It would be beneficial to mention achievements of other research groups regarding cellulose hydrolysis using ionic liquids. Why were these specific ionic liquids chosen? It is advisable to include this information in the introduction.
I don't quite understand the trend of listing everything that was done in the article at the end of the introduction, which some authors use. This is what the abstract is for. It would be better at the end of the introduction to formulate the aim of the study or/and propose hypotheses, rather than repeating the abstract one more time.
Line 100. chloro-butane? Was it 1-chlorobutane or 2-chlorobutane? Check the other reagents for excessive hyphens. What is resin cellulose? What was the origin of sugarcane bagasse? How was cellulose extracted?
Line 142. Please check why the origin of common IL is “finally” after the description of the synthesis.
Line 156. Please clarify what means “pretreatment with 100 times of stirring for 30 min”
Lines 186-191. Is it supposed to be 2 sentences? Was a fullstop missed somewhere?
Line 275. β
Line 295-296. Check the sentence. Something is missed.
Line 303. Well, there are other reasons to use sulfuric acid than environmental and personnel friendliness. The authors write “compared”, but in fact, no comparison has yet been given, except for the environmental friendliness of ionic liquids vs sulfuric acid. Yield, time, amount of reagents, electricity, financial costs were not compared. Only after a complete study of all parameters, as well as scaling (if at all possible) of the technology, the authors can talk about “urgent measures”. Now it is still premature.
Lines 312-313. The sentence repeats for the third time in this paragraph that temperature influences the reaction of hydrolysis. One time is enough, people who read this journal know this fact.
Line 318. Consider changing the confusing phrase “on the other hand” since there is no contradiction with the previous sentence.
Line 319. ionized hydrogen ions?
Line 322, 329. hydrolysis law?
Line 363-365. “The results are shown in Table 2. It can be seen that these four ionic liquids have no significant pretreatment effect on sugarcane cellulose compared to [Bmim]Cl”. It can not be seen from table 2 because [Bmim]Cl is absent there.
Lines 365-366. “Cl- ionic liquids is better than that with Br- ionic liquids” How the authors can prove that it was the anion’s influence? There is no IL with similar cations and different anions in Table 2.
Line 369 and others. Please use consistency in the spelling of
pre-treatment/pretreatment in the whole text.
Line 382. Cellulose has infrared structure??? Is it really what FTIR characterizes?
Figures 6-8. Despite defining the samples in the text, please mention the samples differences in the footnote of the figures. Returning to the advice about the experimental design figure, it would be good to mark on it exactly which samples were studied in this section.
Line 391. “It can be seen that the spectrum c is almost consistent with that of the original sugarcane cellulose sample”. Can it be seen? As far as I have understood, there is no spectrum of original sugarcane cellulose in figure 6.
Please paraphrase the part “indicating that sugarcane cellulose is directly hydrolyzed with water without ionic liquid pretreatment and under the conditions of our reaction investigation”. It is confusing to read that cellulose was hydrolized with water and then, later, about some “our conditions”. I managed to understand what the authors mean, but after the third attempt, taking into account the length of this sentence. It is just about the clarity of the sentence. More precise mentioning of “our conditions” will also benefit the clarity.
Lines 395-398. “the peak at 1428 cm-1 is attributed to the shear movement of -CH2. <...> the intensity of this peak <…> is significantly weakened, indicating that The intramolecular hydrogen bond with sugarcane cellulose O-6 is destroyed”. Please describe in the text how the adsorption band at 1428 cm-1 is related to hydrogen bonding? What is cellulose O-6?
Line 402. Determine in what region of the spectra are situated “O-H vibration peaks”.
Line 457. “As the temperature increases, the viscosity of the system decreases”. Please clarify what the authors mean by viscosity of the system. As far as I understood, there are solid cellulose particles, liquid “melted” IL, and water. Viscosity of what component decreases? Of IL?
Line 465. Figure11 should be figure 9.
Line 486. Check the sentence. Something went wrong.
Line 561. “there was no new peak in sugarcane cellulose after hydrolysis”. Did the authors expect the appearance of any new peak?
Please add doi to the references.
Comments on the Quality of English LanguageThe English needs improvement. The authors use overly long and confusing sentence constructions. An example is the sentence in lines 419-424, that is impossible to comprehend even after the third attempt: “In the SEM observation, it can be found that sample b shows a small number of SEM images similar to sample A, that is, the damaged massive structure, but most of the structure; as shown in Figure 7(1b) and (2b), a relatively complete fiber structure is maintained, and the surface is relatively smooth, which may be because a small amount of extra water is added after the sugarcane cellulose is pretreated, and the water added in this process makes it partially sweet”.
Additionally, incorrect usage of prepositions (e.g., “the contribution of ionic liquid on the hydrolysis” – line 333, it is not the only one) has been identified.
Author Response
Reviewer 3
Authors are recommended to carefully proofread the text. Punctuation marks and spaces, subscripts in formulae are missing in many sentences.
My review includes questions for the authors. These are not questions to satisfy my own curiosity. These are points in the text that were not clear to me as a reader. Accordingly, I assume that this may not be clear to other readers either. Therefore, if the question has a right to exist, I suggest correcting the text itself, instead of answering me in detail in a response without the text improvement, as some authors do. The purpose of a review is to make the manuscript understandable to the reader.
Response: Thanks for your reviewing! We know that there are still many shortcomings in our work and content, and it is hoped that they can be improved through this revision. Thank you for your support and consideration!
The experimental results obtained by the authors are very extensive and varied. Indeed, many types of pretreatments have been studied: with water, without water, homogeneous and heterogeneous reaction, different temperatures, etc, and this is an advantage of the study. On the other hand, it is complicated for a reader who was not directly involved in the experiment to navigate this “sea”. I strongly recommend making an experimental design figure, either as a graphic abstract or as a figure for the second or third part.
Response: Thanks for your guidance! A newly-prepared figure (as new Figure 2) has been provided for more clear description according to your requirement.
Unfortunately, in the third part, there are no references confirming the authors’ hypotheses.
Response: Thanks for your guidance! Some necessary references have been cited in the sections of the third part, which are used to support our results.
I would recommend stylistically adjusting the abstract. Firstly, remove repetitions of phrases (if it is already stated in one sentence that you obtained reducing sugars from cellulose, there is no need to repeat this in the next one, etc.) and even entire sentences. Secondly, it is not necessary to specify which particular type of ion exchange resins you used; not all readers are familiar with this, it's better to mention it in the Materials and Methods section. Avoid using abbreviations, especially without clarification (TBS). In my opinion, it's better to use names rather than formulas of ionic liquids in the abstract. Since this is a research article, some resulting numbers in the abstract are encouraged.
Response: Thanks for your guidance! The abstract has been rewritten according to your requirement.
Lines 34-35. The sentence need improvement. Large hydrogen bond? What is hydrolysis effect of cellulose? Is it an effect?
Response: Sorry for inappropriate expression! They have been adjusted to “large amount of H-bonds” and “hydrolysis result of cellulose”.
Line 37. I do not see description of advantages and disadvantages of these methods above. I am not saying that they are necessary - just the fact that they are not discussed.
Response: Sorry for inappropriate expression! It has been revised as “They all have their own inherent advantages and disadvantages, and exploring mild, green, and efficient degradation technologies has always been a common goal in both academia and industry”.
Line 60. “there are few related reports”. The references are missed.
Response: Thanks for your guidance! Related reference has been cited.
Check the text in line 65. Something is wrong with this sentence.
Response: Thanks for your guidance! It has been revised.
Line 69. “the base group of bagasse is divided into fibrin, hemicellulose and lignin”. What mean the base group? Also, check the information about component content of biomass.
Response: Sorry for these problems! It should be basic components. That is, the basic components of bagasse include cellulose, hemicelluloses and lignin.
The introduction lacks justification for why ionic liquids are suitable for bagasse hydrolysis. The authors have described the drawbacks of acidic and alkaline hydrolysis. Therefore, there is no choice but to resort to hydrolysis using ionic liquids? This was the motivation? It would be beneficial to mention achievements of other research groups regarding cellulose hydrolysis using ionic liquids. Why were these specific ionic liquids chosen? It is advisable to include this information in the introduction.
Response: Thanks for your guidance! Recent references about the ILs and cellulose have been cited and discussed, and necessary elucidation has also been added in the Section of Introduction.
I don't quite understand the trend of listing everything that was done in the article at the end of the introduction, which some authors use. This is what the abstract is for. It would be better at the end of the introduction to formulate the aim of the study or/and propose hypotheses, rather than repeating the abstract one more time.
Response: Thanks for your guidance! The end of the introduction has been reorganized rather than repeating the abstract.
Line 100. chloro-butane? Was it 1-chlorobutane or 2-chlorobutane? Check the other reagents for excessive hyphens. What is resin cellulose? What was the origin of sugarcane bagasse? How was cellulose extracted?
Response: Sorry for these problems! It should be 1-chlorobutane. Other excessive hyphens have been checked and removed. Resin cellulose has been revised as resin. Sugarcane bagasse was collected from the local sugarcane market. The raw material of sugarcane bagasse (20 mesh) was mixed with distilled water in a solid-liquid ratio of 1:20 (W/V); after stirring for 3 h at 60℃ and filtering, the raw material was dried at 80℃ and ground to 40~100 mesh. Then the powders were mixed with 7% NaOH aqueous solution in a solid-liquid ratio of 1:40 (W/V). After stirring at 70℃ for 8 h and filtering, the residue was collected and washed with distilled water to neutral to obtain sugarcane cellulose, which was dried for further use. These contents have been added in Section 2.1 and 2.3.
Line 142. Please check why the origin of common IL is “finally” after the description of the synthesis.
Response: Sorry, it has been checked.
Line 156. Please clarify what means “pretreatment with 100 times of stirring for 30 min”
Response: Sorry, it should be 100℃.
Lines 186-191. Is it supposed to be 2 sentences? Was a fullstop missed somewhere?
Response: Sorry, it has been revised.
Line 275. Β
Response: Sorry, it has been revised.
Line 295-296. Check the sentence. Something is missed.
Response: Sorry, it has been revised.
Line 303. Well, there are other reasons to use sulfuric acid than environmental and personnel friendliness. The authors write “compared”, but in fact, no comparison has yet been given, except for the environmental friendliness of ionic liquids vs sulfuric acid. Yield, time, amount of reagents, electricity, financial costs were not compared. Only after a complete study of all parameters, as well as scaling (if at all possible) of the technology, the authors can talk about “urgent measures”. Now it is still premature.
Response: Sorry for inappropriate expression! The environmental friendliness of ionic liquids is better than that of sulfuric acid. The hydrolysis time is also shorter. At the same time, the cost of ionic liquids is higher than that of traditional inorganic acids with the same volume, but they can be recycled as green solvents, making them more environmentally friendly, safe, and healthy for operators. There are also many successful industrial cases now, proving the economic feasibility of their actual application. At last, the expression of “urgent measures” has been deleted.
Lines 312-313. The sentence repeats for the third time in this paragraph that temperature influences the reaction of hydrolysis. One time is enough, people who read this journal know this fact.
Response: Thanks for your guidance! It has been deleted.
Line 318. Consider changing the confusing phrase “on the other hand” since there is no contradiction with the previous sentence.
Response: Thanks for your guidance! It has been deleted.
Line 319. ionized hydrogen ions?
Response: Thanks for your guidance! It has been deleted.
Line 322, 329. hydrolysis law?
Response: Thanks for your guidance! It has been replaced by “hydrolysis behaviors”.
Line 363-365. “The results are shown in Table 2. It can be seen that these four ionic liquids have no significant pretreatment effect on sugarcane cellulose compared to [Bmim]Cl”. It can not be seen from table 2 because [Bmim]Cl is absent there.
Response: Sorry for this situation! [Bmim]Cl is truly not included in Table 2, and its result has been introduced in Section 3.1. Now it has been emphasized after revision.
Lines 365-366. “Cl- ionic liquids is better than that with Br- ionic liquids” How the authors can prove that it was the anion’s influence? There is no IL with similar cations and different anions in Table 2.
Line 369 and others. Please use consistency in the spelling ofpre-treatment/pretreatment in the whole text.
Response: Sorry, it has been uniformed.
Line 382. Cellulose has infrared structure??? Is it really what FTIR characterizes?
Response: Sorry for this problem! “infrared” has been removed from this sentence.
Figures 6-8. Despite defining the samples in the text, please mention the samples differences in the footnote of the figures. Returning to the advice about the experimental design figure, it would be good to mark on it exactly which samples were studied in this section.
Response: Thanks for your guidance! It has been marked in the captions.
Line 391. “It can be seen that the spectrum c is almost consistent with that of the original sugarcane cellulose sample”. Can it be seen? As far as I have understood, there is no in figure 6.
Response: Sorry, the spectrum of original sugarcane cellulose has been added in this figure.
Please paraphrase the part “indicating that sugarcane cellulose is directly hydrolyzed with water without ionic liquid pretreatment and under the conditions of our reaction investigation”. It is confusing to read that cellulose was hydrolized with water and then, later, about some “our conditions”. I managed to understand what the authors mean, but after the third attempt, taking into account the length of this sentence. It is just about the clarity of the sentence. More precise mentioning of “our conditions” will also benefit the clarity.
Response: Sorry for unclear expression and poor English! It has been changed to “indicating that the untreatedsugarcane cellulose does not undergo obvious chemical reaction in pure water”.
Lines 395-398. “the peak at 1428 cm-1 is attributed to the shear movement of -CH2. <...> the intensity of this peak <…> is significantly weakened, indicating that The intramolecular hydrogen bond with sugarcane cellulose O-6 is destroyed”. Please describe in the text how the adsorption band at 1428 cm-1 is related to hydrogen bonding? What is cellulose O-6?
Response: Thanks for your guidance! In cellulose molecular chains, a large amount of hydrogen bonds exist. A part of hydrogen bonds connect the O-6 atom (oxygen at position 6) in the chain to O-2 'and O-3 to O-5', forming the polymer chains; after being inserted into the lattice, O-6 on one chain can also form hydrogen bonds with O-3 on adjacent chains, forming the network structure of cellulose. The mentioned -CH2 is connected to O-6, when the O-6 involved hydrogen bonds are destroyed, the vibration of -CH2will be affected [21]. The following scheme has been added in the new Figure 2 to make readers clear.
Line 402. Determine in what region of the spectra are situated “O-H vibration peaks”.
Response: Sorry for the negligence! The range is 1600~1650 cm-1.
Line 457. “As the temperature increases, the viscosity of the system decreases”. Please clarify what the authors mean by viscosity of the system. As far as I understood, there are solid cellulose particles, liquid “melted” IL, and water. Viscosity of what component decreases? Of IL?
Response: Sorry for unclear expression! It refers to the solution system, which is the IL aqueous solution.
Line 465. Figure11 should be figure 9.
Response: Sorry, it has been revised.
Line 486. Check the sentence. Something went wrong.
Response: Sorry, it has been revised.
Line 561. “there was no new peak in sugarcane cellulose after hydrolysis”. Did the authors expect the appearance of any new peak?
Response: No. The results indicated that there was no derivatization reaction between sugarcane cellulose and ionic liquid during the hydrolysis process.
Please add doi to the references.
Response: Sorry, it has been added.
Round 2
Reviewer 1 Report
Comments and Suggestions for Authors
Unfortunately, I still found the manuscript quite chaotic. Furthermore, although the authors described other related works in the introduction, they failed to relate their results to the state of the art. Additionally, it is still not clear why the authors chose these two ionic liquids for the study.
Author Response
Unfortunately, I still found the manuscript quite chaotic. Furthermore, although the authors described other related works in the introduction, they failed to relate their results to the state of the art. Additionally, it is still not clear why the authors chose these two ionic liquids for the study.
Dear reviewer
We list those ionic liquids involved in different catalytic systems as follows to provide a clearer description of our selection about the two ionic liquids.
(1) If co catalysts are not intended, the acidity of ionic liquids should be as strong as possible;
(2) Benzothiazolium ionic liquids exhibit advantages over imidazolium ILs in the catalytic process of homogeneous/heterogeneous switching;
(3) If the heterogeneous catalytic performance is expected to be better, both the anion and cation had better have acidic groups, so that the acidity is higher;
(4) The ionic liquids that have already been applied in this field are not intended for investigation in this study, because we want to find more other efficient ILs.
Due to the above reasons and comprehensive considerations, combined with the screening results of the preliminary experiment, the ILs of [HBth][CH3SO3] and [C3SO3Hmim]HSO4 were finally selected as the tested catalysts in our experiments.
IL cation |
IL anion |
cocatalyst |
catalysis system |
ref |
[Bmim] |
[Cl] |
H2SO4 |
homogeneous |
[19] |
[Bmim] |
[Cl] |
DuPontTM AmberLystTM 15DRY polymeric catalyst |
heterogeneous |
[20] |
[HSO3Bmim] |
[HSO4] |
MnCl2 |
homogeneous |
[21] |
[HBth] |
[p-Tsa] |
- |
homogeneous at high temperature for efficient reaction, and heterogeneous at low temperature for easy post-treatment |
[32] |
[Pmim] |
[Cl] |
SO3H-biochar |
heterogeneous |
[33] |
[HBth] |
[CH3SO3] |
- |
homogeneous |
this study |
[C3SO3Hmim] |
[HSO4] |
- |
heterogeneous |
Reviewer 2 Report
Comments and Suggestions for Authors
None
Author Response
Thanks for your review.
Reviewer 3 Report
Comments and Suggestions for Authors
The authors have improved the manuscript greatly. However, in lines 12-13, the sentence is still duplicated.
Author Response
The authors have improved the manuscript greatly. However, in lines 12-13, the sentence is still duplicated.
response:Thanks for guidance! The problem has been checked and revised.